# On Episodes, Prototypical Networks, and Few-Shot Learning

**Steinar Laenen** *
School of Informatics
University of Edinburgh
`V.S.E.Laenen@sms.ed.ac.uk`

**Luca Bertinetto**
FiveAI
`luca.bertinetto@five.ai`

## Abstract

Episodic learning is a popular practice among researchers and practitioners interested in few-shot learning. It consists of organising training in a series of learning problems (or *episodes*), each divided into a small training and validation subset to mimic the circumstances encountered during evaluation. But is this always necessary? In this paper, we investigate the usefulness of episodic learning in methods which use nonparametric approaches, such as nearest neighbours, at the level of the episode. For these methods, we not only show how the constraints imposed by episodic learning are not necessary, but that they in fact lead to a data-inefficient way of exploiting training batches. We conduct a wide range of ablative experiments with Matching and Prototypical Networks, two of the most popular methods that use nonparametric approaches at the level of the episode. Their "non-episodic" counterparts are considerably simpler, have less hyperparameters, and improve their performance in multiple few-shot classification datasets.

## 1   Introduction

The problem of few-shot learning (FSL) – classifying examples from previously unseen classes given only a handful of training data – has considerably grown in popularity within the machine learning community in the last few years. The reason is likely twofold. First, being able to perform well on FSL problems is important for several applications, from learning new symbols [23] to drug discovery [2]. Second, since the aim of researchers interested in meta-learning is to design systems that can quickly learn novel concepts by generalising from previously encountered learning tasks, FSL benchmarks are often adopted as a practical way to empirically validate meta-learning algorithms.

To the best of our knowledge, there is not a widely recognised definition of meta-learning. In a recent survey, Hospedales et al. [22] informally describe it as *"the process of improving a learning algorithm over multiple learning episodes"*. In practical terms, following the compelling rationale that *"test and train conditions should match"* [48, 13], several seminal meta-learning papers (e.g. [48, 32, 14]) have emphasised the importance of organising training into *episodes*, i.e. learning problems with a limited amount of "training" (the *support* set) and "test" examples (the *query* set) to mimic the test-time scenario presented by FSL benchmarks.

However, several recent works (e.g. [9, 49, 11, 44]) showed that simple baselines can outperform established FSL meta-learning methods by using embeddings pre-trained with the standard cross-entropy loss, thus casting a doubt on the importance of episodes in FSL. Inspired by these results, we aim at understanding the practical usefulness of episodic learning in popular FSL methods relying on metric-based nonparametric classifiers such as Matching and Prototypical Networks [48, 40]. We chose this family of methods because they do not perform any adaptation at test time. This allows us

---

*Work done while research intern at FiveAI

35th Conference on Neural Information Processing Systems (NeurIPS 2021).

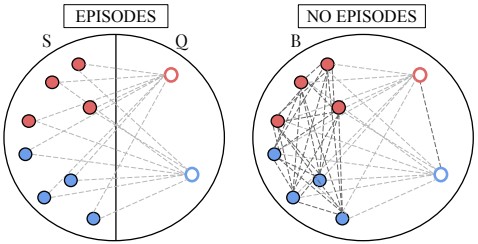

| | POSITIVES | NEGATIVES |
|---|---|---|
| NO EPISODES | $\binom{m+n}{2}w$ | $\binom{w}{2}(m+n)^2$ |
| EPISODES | $wmn$ | $w(w-1)mn$ |
| PAIRS LOST | $\frac{w}{2}(m^2+n^2-m-n)$ | $\frac{w}{2}(w-1)(m^2+n^2)$ |

Figure 1: Difference in *batch exploitation* for metric-based methods between adopting or not adopting the concept of *episodes* during training, on an illustrative few-shot learning problem with 2 *ways* (classes), and 4 *shots* (examples) and 1 *query* per class.

Table 1: The extra number of gradients that, on the *same batch*, a non-episodic method can exploit with respect to its episodic counterpart grows quadratically as $O(w^2(m^2+n^2))$, where $w$ is the number of ways, and $n$ and $m$ are the number of shots and queries per class.

to test the efficacy of episodic training without having to significantly change the baseline algorithms, which could potentially introduce confounding factors.

In this work we perform a case study focussed on Matching Networks [48] and Prototypical Networks [40], and we show that within this family of methods episodic learning *a)* is detrimental for performance, *b)* is analogous to randomly discarding examples from a batch and *c)* introduces a set of superfluous hyperparameters that require careful tuning. Without episodic learning, these methods are closely related to the classic *Neighbourhood Component Analysis* (NCA) [19, 35] on deep embeddings and achieve, without bells and whistles, an accuracy that is competitive with recent methods on multiple FSL benchmarks: *mini*ImageNet, CIFAR-FS and *tiered*ImageNet.

PyTorch code is available at `https://github.com/fiveai/on-episodes-fsl`.

## 2 Background and method

This section is divided as follows: Sec. 2.1 introduces episodic learning and illustrates a data efficiency issue encountered with nonparametric few-shot learners based on episodes; Sec. 2.2 introduces the losses from Snell et al. [40], Vinyals et al. [48] and Goldberger et al. [19] which we use throughout our experiments; and Sec. 2.3 explains the three options we explored to perform FSL classification with previously-trained feature embeddings.

### 2.1 Episodic learning

A common strategy to train FSL methods is to consider a distribution $\hat{\mathcal{E}}$ over possible subsets of labels that is as close as possible to the one encountered during evaluation $\mathcal{E}$ [2] [48]. Each *episodic batch* $B_E = \{S, Q\}$ is obtained by first sampling a subset of labels $L$ from $\hat{\mathcal{E}}$, and then sampling images constituting both *support set $S$* and *query set $Q$* from the set of images with labels in $L$, where $S = \{(\mathbf{s}_1, y_1), \ldots, (\mathbf{s}_n, y_n)\}$, $Q = \{(\mathbf{q}_1, y_1), \ldots, (\mathbf{q}_m, y_m)\}$, and $S_k$ and $Q_k$ denote the sets of images with label $y = k$ in the support set and query set respectively.

For most methods, this corresponds to training on a series of mini-batches in which each image belongs to *either* the support *or* the query set. Support and query sets are constructed such that they both contain all the classes of $L$, and a fixed number of images per class. Therefore, episodes are defined by three variables: the number of classes $w = |L|$ (the "ways"), the number of examples per class in the support set $n = |S_k|$ (the "shots"), and the number of examples per class in the query set $m = |Q_k|$. During evaluation, the set $\{w, n, m\}$ defines the problem setup. Instead, at training time $\{w, n, m\}$ can be seen as a set of hyperparameters controlling the batch creation, and that (as we will see in Sec. 3.2) requires careful tuning.

---

[2]Note that, in FSL, the sets of classes encountered during training and evaluation are disjoint.

In a Maximum Likelihood Estimation framework, training on these episodes can be written as

$$\arg\max_{\theta} \ \mathbb{E}_{L \sim \hat{\mathcal{E}}} \ \mathbb{E}_{\substack{S \sim L \\ Q \sim L}} \left( \sum_{(q_i, y_i) \in Q} \log P_\theta \left( y_i | q_i, S, \rho \right) \right). \tag{1}$$

For the sake of brevity, with a slight abuse of notation we omit the function $f_\theta$ (e.g. a deep neural network) which is used to obtain a representation for the images in $S$ and $Q$, and whose parameters $\theta$ are optimised during the training process. Note that the optimisation of Eq. 1 depends on an *optional* set of parameters $\rho$. This is obtained by an "inner" optimisation procedure, whose scope is limited to the current episode [22]. The idea is that the "outer" optimisation loop, by attending to a distribution of episodes, will appropriately shift the *inductive bias* of the algorithm located in the inner loop, thus learning how to learn [47]. In recent years, many interesting proposals have been made about what form $\rho$ should have, and how it should be computed. For instance, in MAML [14] $\rho$ takes the form of an update of the global parameters $\theta$, while Ravi and Larochelle [32] learn to optimise by considering $\rho$ as set of the hyper-parameters of the optimiser's update rule.

Other methods, such as Matching and Prototypical Networks [48, 40], avoid learning a separate set of parameters $\rho$ altogether, and utilise a nonparametric learner (such as nearest neighbour classifiers) at the inner level. We chose to focus our case study on these methods not only because they have been seminal for the community, but also for ease of experimental design. Having $\rho = \varnothing$ considerably reduces the design complexity of the algorithm, thus allowing precise ablations to understand the efficacy of episodic learning without considerably changing the nature of the original algorithms.

**Considerations on data efficiency.** The constraints imposed by episodic learning on the role each image has in a training batch has subtle but important implications, illustrated in Fig. 1 by highlighting the number of distances contributing to the loss. By dividing batches between support and query set ($S$ and $Q$) during training, *episodes* have the side effect of disregarding many of the distances between labelled examples that would constitute useful training signal for nonparametric FSL methods. More specifically, for metric-based nonparametric methods, the number of training distances that are omitted in a batch because of the episodic strategy grows quadratically as $O(w^2(m^2 + n^2))$ (derivation shown in Appendix A). Table 1 breaks down this difference in terms of gradients from positives and negatives distance pairs (which we simply refer to as *positives* and *negatives* throughout the rest of the paper). In a typical training batch with $w = 20$, $m = 15$ and $n = 5$ [40], ignoring the episodic constraints increases the number of both positives and negatives by more than 150%.

In the remainder of this paper, we conduct a case study to illustrate how this issue affects two of the most popular FSL algorithms relying on nonparametric approaches at the inner level: Prototypical Networks [40] and Matching Networks [48].

## 2.2  Loss functions

**Prototypical Networks (PNs)** [40] are one of the most popular and effective approaches in the few-shot learning literature. They are at the core of several recently proposed FSL methods (e.g. [27, 18, 1, 50, 7]), and they are used in a number of applied machine learning works (e.g. EEG scan analysis for autism [36] and glaucoma grading [17]).

During *training*, episodes consisting of a support set $S$ and a query set $Q$ are sampled as described in Sec. 2.1. Then, a *prototype* for each class $k$ is computed as the mean embedding of the samples from the support set belonging to that class: $\mathbf{c}_k = (1/|S_k|) \cdot \sum_{(\mathbf{s}_i, y_k) \in S_k} f_\theta(\mathbf{s}_i)$, where $f_\theta$ is a deep neural network with parameters $\theta$ learned via Eq. 1.

Let $C = \{(\mathbf{c}_1, y_1), \dots, (\mathbf{c}_k, y_k)\}$ be the set of prototypes and corresponding labels. The loss can be written as follows:

$$\mathcal{L}_{\text{PNs}} = \frac{-1}{|Q|} \sum_{(\mathbf{q}_i, y_i) \in Q} \log \left( \frac{\exp -\|f_\theta(\mathbf{q}_i) - \mathbf{c}_{y_i}\|^2}{\sum_{k'} \exp -\|f_\theta(\mathbf{q}_i) - \mathbf{c}_{k'}\|^2} \right),$$

where $k'$ is an index that goes over all classes.

**Matching Networks (MNs)** [48] are closely related to PNs in the multi-shot case and equivalent in the 1-shot case. Rather than aggregating the embeddings of the same class into prototypes, this loss directly computes a softmax over individual embeddings of the support set, as:

$$\mathcal{L}_{\text{MNs}} = \frac{-1}{|Q|} \sum_{\substack{(\mathbf{q}_i, y) \\ \in Q}} \log \left( \frac{\sum_{\substack{\mathbf{s}_j \\ \in S_y}} \exp -\|f_\theta(\mathbf{q}_i) - f_\theta(\mathbf{s}_j)\|^2}{\sum_{\substack{\mathbf{s}_k \\ \in S}} \exp -\|f_\theta(\mathbf{q}_i) - f_\theta(\mathbf{s}_k)\|^2} \right).$$

In their work, Vinyals et al. [48] use the cosine rather than the Euclidean distance. However, (as [40]) we observed that the Euclidean distance is a better choice for FSL problems, and thus we use it in all the losses of our experiments. Note that Vinyals et al. [48] also suggest a variant to $\mathcal{L}_{\text{MNs}}$ (MNs with "Full Context Embeddings"), where an LSTM (with an extra set of parameters) is used to condition the way the inputs are embedded in the current support set. In our experiments, we did not consider this variant as it falls in the category of *adaptive* episodic learning approaches ($\rho \neq \varnothing$, see Sec. 2.1).

**Neighbourhood Component Analysis (NCA).** $\mathcal{L}_{\text{MNs}}$ and $\mathcal{L}_{\text{PNs}}$ sum over the likelihoods that a query image belongs to the same class of a certain sample (or prototype) from the support set by computing the softmax over the distances between the query and the support samples (or prototypes). This is closely related to the *Neighbourhood Component Analysis* approach by Goldberger et al. [19] (and expanded to the non-linear case by Salakhutdinov et al. [35] and Frosst et al. [15]), except for a few important differences which we discuss at the end of this section.

Let $i \in [1, b]$ be the indices of the images within a batch $B$. The NCA loss can be written as:

$$\mathcal{L}_{\text{NCA}} = \frac{-1}{|B|} \sum_{i \in 1,\ldots,b} \log \left( \frac{\sum_{\substack{j \in 1,\ldots,b \\ j \neq i \\ y_i = y_j}} \exp -\|\mathbf{z}_i - \mathbf{z}_j\|^2}{\sum_{\substack{k \in 1,\ldots,b \\ k \neq i}} \exp -\|\mathbf{z}_i - \mathbf{z}_k\|^2} \right),$$

where $\mathbf{z}_i = f_\theta(\mathbf{x}_i)$ is an image embedding and $y_i$ its corresponding label. By minimising this loss, distances between embeddings from the same class will be minimised, while distances between embeddings from different classes will be maximised. Importantly, note how the concepts of support set and query set here do not exist. More simply, the images (and respective labels) constituting the batch $B = \{(\mathbf{x}_1, y_1), \ldots, (\mathbf{x}_b, y_b)\}$ are sampled uniformly.

Given the similarity between these three losses, and considering that PNs and MNs do not perform episode-specific parameter adaptation, $\{w, m, n\}$ can be simply interpreted as the set of hyperparameters controlling the sampling of mini-batches during training. More specifically, PNs, MNs and NCA differ in three aspects:

I. First and foremost, due to the nature of episodic learning, PNs and MNs only consider pairwise distances between the query and the support set (Fig. 1 left); NCA instead uses *all* the distances within a batch and treats each example in the same way (Fig. 1 right).
II. Only PNs rely on the creation of prototypes.
III. Because of how $L$, $S$ and $Q$ are sampled in episodic learning (Eq. 1), for PNs and MNs some images might be sampled more frequently than others (sampling "with replacement"). NCA instead visits every image of the dataset once for each epoch (sampling "without replacement").

To investigate the effects of these three differences, in Sec. 3 we conduct a wide range of experiments.

## 2.3 Few-shot classification during evaluation

Once $f_\theta$ has been trained, there are many possible ways to perform few-shot classification during evaluation. In this paper we consider three simple approaches that are particularly intuitive for embeddings learned via metric-based losses like the ones described in Sec. 2.2. Note that, in the 1-shot case, all the evaluation methods considered coincide.

$k$**-NN.** To classify an image $\mathbf{q}_i \in Q$, we first compute the Euclidean distance to each support point $\mathbf{s}_j \in S$: $d_{ij} = \|f_\theta(\mathbf{q}_i) - f_\theta(\mathbf{s}_j))\|^2$. Then, we simply assign $y(\mathbf{q}_i)$ to be majority label of the $k$ nearest neighbours. A downside here is that $k$ is a hyper-parameter that has to be chosen, although a reasonable choice in the FSL setup is to set it equal to the number of "shots" $n$.

**Nearest centroid.** Similar to $k$-NN, we can perform classification by inheriting the label of the closest class centroid, i.e. $y(\mathbf{q}_i) = \arg\min_{j \in \{1,...,k\}} \|f_\theta(\mathbf{x}_i) - \mathbf{c}_j\|$. This is the approach used by Prototypical Networks [40], SimpleShot [49], and both baselines of Chen et al. [10].

**Soft assignments.** To classify an image $\mathbf{q}_i \in Q$, we compute the values

$$p_{ij} = \frac{\exp(-\|f_\theta(\mathbf{q}_i) - f_\theta(\mathbf{s}_j))\|^2)}{\sum_{\mathbf{s}_k \in S} \exp(-\|f_\theta(\mathbf{q}_i) - f_\theta(\mathbf{s}_k)\|^2)}$$

for all $\mathbf{s}_j \in S$, which is the probability that $i$ inherits its class from $j$. We then compute the likelihood for each class $k$: $\sum_{s_j \in S_k} p_{ij}$, and choose the class with the highest likelihood $y(\mathbf{q}_i) = \arg\max_k \sum_{s_j \in S_k} p_{ij}$. This is the approach for classification adopted by the original NCA paper [19] and Matching Networks [48].

We experiment with all three alternatives and observe that the *nearest centroid* approach is the most effective (details available in Appendix D). For this reason, unless differently specified, we use it as default in our experiments.

## 3 Experiments

In the following, Sec. 3.1 describes our experimental setup; Sec. 3.2 shows the important effect of the hyperparameters controlling the creation of episodes; in Sec. 3.3 we compare the episodic strategy to randomly discarding pairwise distances within a batch; in Sec. 3.4 we perform a set of ablations to better illustrate the relationship between PNs, MNs and NCA; finally, in Sec. 3.5 we compare our version of the NCA to several recent methods.

### 3.1 Experimental setup

We conduct our experiments on *mini*ImageNet [48], CIFAR-FS [5] and *tiered*ImageNet [34], using the popular ResNet-12 variant first adopted by Lee et al. [24] as embedding function $f_\theta$ [3] . A detailed description of benchmarks, architecture and choice of hyperparameters is deferred to Appendix F, while below we discuss the most important choices of the experimental setup.

Like Wang et al. [49], for all our experiments (including those with Prototypical and Matching Networks) we centre and normalise the feature embeddings before performing classification, as it is considerably beneficial for performance. After training, we compute the mean feature vectors of all the images in the training set: $\bar{\mathbf{x}} = \frac{1}{|\mathcal{D}^{\text{train}}|} \sum_{\mathbf{x} \in \mathcal{D}^{train}} \mathbf{x}$. Then, all feature vectors in the test set are updated as $\mathbf{x}_i \leftarrow \mathbf{x}_i - \bar{\mathbf{x}}$, and normalised by $\mathbf{x}_i \leftarrow \frac{\mathbf{x}_i}{\|\mathbf{x}_i\|}$.

As standard [22], performance is assessed on episodes of 5-way, 15-query and 1- or 5-shot. Each model is evaluated on 10,000 episodes sampled from the validation set during training, or from the test set during testing. To further reduce the variance, we trained each model *three times* with three different random seeds, for a total of 30,000 episodes per configuration, from which 95% confidence intervals are computed.

### 3.2 Batch size and episodes

Despite Prototypical and Matching Networks being among the simplest FSL methods, the creation of episodes requires the use of several hyperparameters ($\{w, m, n\}$, Sec. 2.1) which can significantly affect performance. Snell et al. [40] state that the number of shots $n$ between training and testing should match and that one should use a higher number of ways $w$ during training. In their experiments, they train 1-shot models with $w = 30$, $n = 1$, $m = 15$ and 5-shot models with $w = 20$, $n = 5$, $m = 15$, with batch sizes of 480 and 400, respectively. Since the corresponding batch sizes of these configurations differ, making direct comparisons between them is difficult.

---

[3]Note that, since we do not use a final linear layer for classification, our backbone is in fact a ResNet-11.

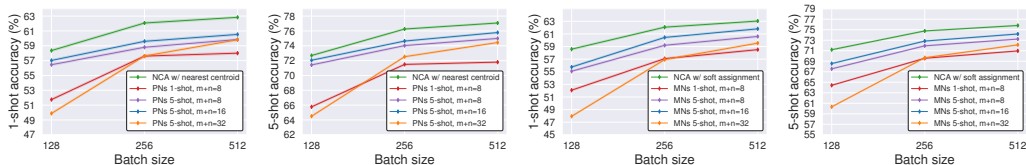

Figure 2: 1-shot and 5-shot accuracies on CIFAR-FS (val. set) for Prototypical and Matching Networks models trained with different episodic configurations: 1-shot with $m+n=8$ and 5-shot with $m+n=8$, 16 or 32. NCA models are trained on batches of size 128, 256 and 512 to match the size of the episodes. Reported values correspond to the mean accuracy of three models trained with different random seeds and shaded areas represent 95% confidence intervals. See Sec. 3.2 for details.

Instead, to directly compare configurations across batch sizes, we define an episode by its number of shots $n$, the batch size $b$ and the total number of images per class $m+n$ (the sum of elements across support and query set). For example, if we train a 5-shot model with $m+n=8$ and $b=256$, its corresponding training episodes will have $n=5$, $m=8-n=3$, and $w=256/(m+n)=32$. Using this notation, we train configurations of PNs and MNs covering several combinations of these hyperparameters, so that the resulting batch size corresponding to an episode is 128, 256 or 512. Then, we train three configurations of the NCA, where the sole hyperparameter is the batch size $b$.

Results for CIFAR-FS can be found in Fig. 2, where we report results for NCA, PNs and MNs with $m+n=8$, 16 or 32. Results for *mini*ImageNet observe the same trend and are deferred to Appendix H. For consistency in our comparisons, we evaluate performance using a *nearest centroid* classifier when comparing against PNs, and *soft assignments* when comparing against MNs (see Sec. 2.3). Note that PNs and MNs results for 1-shot with $m+n=16$ and $m+n=32$ are not reported, as they fare significantly worse. The 1-shot $m+n=16$ is 4% worse in the best case compared to the lowest lines in Fig. 2, and the $m+n=32$ is 10% worse in the best case. This is likely because these two setups exploit the fewest number of pairs among all the setups, which leads to the least training signal being available. In Appendix E we discuss whether the difference in performance between the different episodic batch setups of Fig. 2 can be solely explained by the differences in the number of distance pairs used in the batch configurations. We indeed find that generally speaking the higher the number of pairs the better. However, one should also consider the positive/negative balance and the number of classes present within a batch.

Several things can be observed from Fig. 2. First, NCA-trained embeddings perform better than *all* configurations, no matter the batch size. Second, PNs and MNs are very sensitive to different hyperparameter configurations. For instance, with batches of size 128, PNs trained with episodes of 5-shot and $m+n=32$ perform worse than a PNs trained with 5-shot episodes and $m+n=16$. Note that, as we will show in Table 2, the best episodic configurations for PNs and MNs found with this hyperparameter search is superior to the setting used in the original papers.

### 3.3 Episodic batches vs. random sub-sampling

Despite the inferior performance with respect to the NCA, one might posit that, by training on episodes, PNs and MNs can somehow make better use of a smaller number of distances within a batch. This could be useful, for instance, in situations where it is important to train with very large batches. Given the increased conceptual complexity and the extra hyperparameters, the efficacy of episodic learning (in cases where a smaller number of distances should be considered) should be validated against the much simpler approach of random subsampling. We perform an experiment where we train NCA models by randomly discarding a fraction of the total number of distances used in the loss. Then, for comparison, we include different PNs and MNs models, after having computed to which percentage of discarded pairs (in a normal batch) their episodic batches correspond to.

Results can be found in Fig. 3. As expected, we can see how subsampling a fraction of the total available number of pairs within a batch negatively affects performance. More importantly, we can notice that the points representing PNs and MNs models lie very close to the under-sampling version of the NCA. This suggests that the episodic strategy is roughly equivalent, empirically, to only exploiting a fraction of the distances available in a batch. Note how, moving along the x-axis of Fig. 3, variants of PNs and MNs exploiting a higher number of distances perform better.

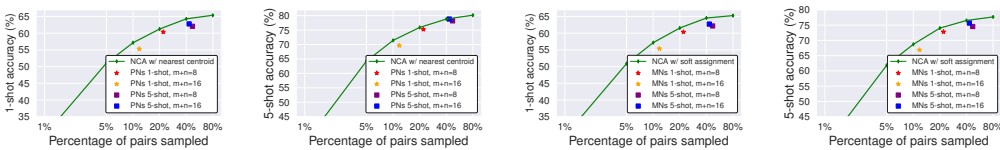

Figure 3: 1-shot and 5-shot accuracies on *mini*ImageNet (val. set) for NCA models trained by only sampling a fraction of the total number of available pairs in the batch (of size 256). Stars and squares represent models trained using the Prototypical or Matching Network loss, and are plotted on the x-axis based on the total number of distance pairs exploited in the loss, so that they can be directly compared with this "sub-sampling" version of the NCA. Reported values correspond to the mean accuracy of three models trained with different random seeds and shaded areas represent 95% confidence intervals. See Sec. 3.3 for details.

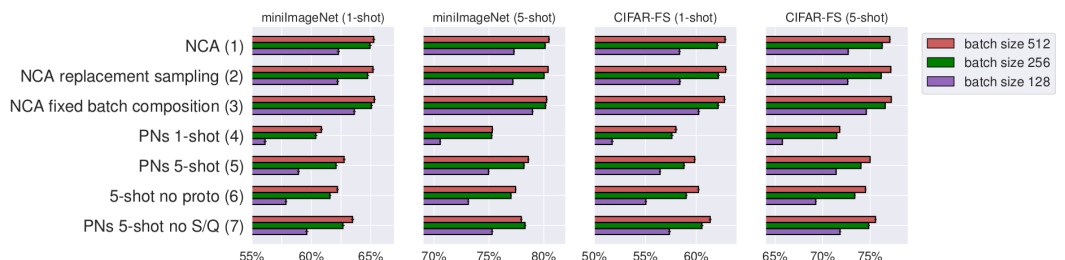

Figure 4: Ablation experiments on NCA and Prototypical Networks, both on batches (or episodes) of size 128, 256, and 512 on *mini*ImageNet and CIFAR-FS (val. set). Reported values correspond to the mean accuracy of three models trained with different random seeds and error bars represent 95% confidence intervals. See Sec. 3.4 for details.

## 3.4 Ablation experiments

To better analyse why NCA performs better, in this section we consider the three key differences discussed at the end of Sec. 2.2 by performing a series of ablations on models trained on batches of size 128, 256 and 512. Results are summarised in Fig. 4. We refer the reader to Appendix B to obtain detailed steps describing how these ablations affect the losses of Sec. 2.2.

First, we compare two variants of the NCA: one in which the sampling of the training batches happens sequentially and without replacement, as is standard in supervised learning, and one where batches are sampled with replacement. This modification (row 1 and 2 of Fig. 4) has a negligible effect, meaning that the replacement sampling introduced by episodic learning should not interfere with the other ablations. We then perform a series of ablations on episodic batches, i.e. sampled with the method described in Sec. 2.1. To obtain a reasonably-performing model for both 1- and 5-shot models, we use configurations with $m + n = 8$. This means that, for PNs and MNs, models are trained with 8 images per class, and either 16, 32 or 64 classes (batches of size 128, 256 and 512 respectively). The batch size for NCA is also set to either 128, 256, or 512, allowing direct comparison.

The ablations of Fig. 4 compare PNs to NCA. First, we train standard PNs models (row 4 and 5 of Fig. 4). Next, we train a model where "prototypes" are not computed (row 6). This implies that, similar to what happens in MNs, distances are considered between individual points, but a separation between query and support set remains. This ablation allows us to investigate if the loss in performance by PNs compared to NCA can be attributed to prototype computation during training (which turned out not to be the case). Then, we perform an ablation where we ignore the separation between support and query set, and compute the NCA on the union of the support and query set, while still computing prototypes for the points that would belong to the support set (row 7). Last, we perform an ablation where we consider all the previous points together: we sample with replacement, we ignore the separation between support and query set and we do not compute prototypes (row 3). This amounts to the NCA loss, except that it is computed on batches with a fixed number of classes and a fixed number of images per class (row 3). Notice that in Fig. 4 there is only one row dedicated to 1-shot models. This is because we cannot generate prototypes from 1-shot models, so we cannot

| | *mini*ImageNet | | CIFAR-FS | | *tiered*ImageNet | |
|---|---|---|---|---|---|---|
| | **1-shot** | **5-shot** | **1-shot** | **5-shot** | **1-shot** | **5-shot** |
| *Episodic methods* | | | | | | |
| adaResNet [26] | $56.88 \pm 0.62$ | $71.94 \pm 0.57$ | - | - | - | - |
| TADAM[27] | $58.50 \pm 0.30$ | $76.70 \pm 0.30$ | - | - | - | - |
| Shot-Free [33] | $60.71\pm$ n/a | $77.64\pm$ n/a | $69.2\pm$ n/a | $84.7\pm$ n/a | $63.52\pm$ n/a | $82.59\pm$ n/a |
| TEAM [30] | $60.07\pm$ n/a | $75.90\pm$ n/a | - | - | - | - |
| MTL [42] | $61.20 \pm 1.80$ | $75.50 \pm 0.80$ | - | - | - | - |
| TapNet [50] | $61.65 \pm 0.15$ | $76.36 \pm 0.10$ | - | - | $63.08 \pm 0.15$ | $80.26 \pm 0.12$ |
| MetaOptNet-SVM[24] | $62.64 \pm 0.61$ | $78.63 \pm 0.46$ | $\mathbf{72.0 \pm 0.7}$ | $84.2 \pm 0.5$ | $65.99 \pm 0.72$ | $81.56 \pm 0.53$ |
| Variatonal FSL [51] | $61.23 \pm 0.26$ | $77.69 \pm 0.17$ | - | - | - | - |
| *Simple cross-entropy baselines* | | | | | | |
| Transductive finetuning [11] | $62.35 \pm 0.66$ | $74.53 \pm 0.54$ | $70.76 \pm 0.74$ | $81.56 \pm 0.53$ | - | - |
| RFIC-simple [44] | $62.02 \pm 0.63$ | $\mathbf{79.64 \pm 0.44}$ | $71.5 \pm 0.8$ | $\mathbf{86.0 \pm 0.5}$ | $\mathbf{69.74 \pm 0.72}$ | $\mathbf{84.41 \pm 0.55}$ |
| Meta-Baseline [10] | $\mathbf{63.17 \pm 0.23}$ | $79.26 \pm 0.17$ | - | - | $68.62 \pm 0.27$ | $83.29 \pm 0.18$ |
| *Our implementations:* | | | | | | |
| MNs ([40] episodes) | $58.91 \pm 0.12$ | $72.48 \pm 0.10$ | $69.28 \pm 0.13$ | $80.79 \pm 0.10$ | $65.75 \pm 0.13$ | $78.40 \pm 0.10$ |
| PNs ([40] episodes) | $59.78 \pm 0.12$ | $75.42 \pm 0.09$ | $69.94 \pm 0.12$ | $84.01 \pm 0.09$ | $65.80 \pm 0.13$ | $81.26 \pm 0.10$ |
| MNs (our episodes) | $60.77 \pm 0.12$ | $73.82 \pm 0.09$ | $71.86 \pm 0.13$ | $82.41 \pm 0.10$ | $66.53 \pm 0.13$ | $79.08 \pm 0.10$ |
| PNs (our episodes) | $61.32 \pm 0.12$ | $77.77 \pm 0.09$ | $70.41 \pm 0.12$ | $84.61 \pm 0.10$ | $66.89 \pm 0.14$ | $82.20 \pm 0.09$ |
| SimpleShot [49] | $62.16 \pm 0.12$ | $78.33 \pm 0.09$ | $69.98 \pm 0.12$ | $84.40 \pm 0.09$ | $66.67 \pm 0.14$ | $81.57 \pm 0.10$ |
| **NCA soft assignment (ours)** | $62.55 \pm 0.12$ | $76.93 \pm 0.11$ | $\mathbf{72.49 \pm 0.12}$ | $83.38 \pm 0.09$ | $68.35 \pm 0.13$ | $81.04 \pm 0.09$ |
| **NCA nearest centroid (ours)** | $62.55 \pm 0.12$ | $78.27 \pm 0.09$ | $\mathbf{72.49 \pm 0.12}$ | $85.15 \pm 0.09$ | $68.35 \pm 0.13$ | $83.20 \pm 0.10$ |

Table 2: Comparison of methods that use ResNet12 as $f_\theta$, on the test set of *mini*ImageNet, CIFAR-FS, and *tiered*ImageNet. Values are reported with 95% confidence intervals. For our methods, reported values correspond to the mean accuracy of three models trained with different random seeds.

have a "no proto" ablation. Furthermore, for 1-shot models the "no S/Q" ablation is equivalent to the NCA with a fixed batch composition.

From Fig. 4, we can see that disabling prototypes (row 6) negatively affects the performance of 5-shot (row 5), albeit slightly. Since for PNs the amount of gradient signal is the same with (row 5, Fig.4) or without (row 6, Fig.4) the computation of prototypes, we believe that this could be motivated by the increased misalignment between the training and test setup present in the ablation of row 6. Nonetheless, enabling the computation between all pairs increases the performance (row 7) and, importantly, enabling *all* the ablations (row 3) completely recovers the performance lost by PNs. Note the meaningful gap in performance between row 1 and 3 in Fig. 4 for batch size 128, which disappears for batch size 512. This is likely due to the number of positives available in an excessively small batch size. Since our vanilla NCA creates batches by simply sampling images randomly from the dataset, there is a limit to how small a batch can be (which depends on the number of classes of the dataset). As an example, consider the extreme case of a batch of size 4. For the datasets considered, it is very likely that such a batch will contain no positive pairs for some classes. Conversely, the NCA ablation with a fixed batch composition (i.e. with a fixed number of images per class) will have a higher number of positive pairs (at the cost of a reduced number of classes per batch). This can explain the difference, as positive pairs constitute a less frequent (and potentially more informative) training signal. In Appendix E we extend this discussion, commenting on the role of positive and negative distances. In Appendix H we also report the results of a second set of ablations to compare NCA and Matching Networks, which are analogous to the ones with Prototypical Networks we just described and lead to the same conclusions.

These experiments highlight that the separation of roles between the images belonging to support and query set, which is typical of episodic learning [48], is detrimental for the performance of metric-based nonparametric few-shot learners. Instead, using the NCA loss on standard mini-batches allows full exploitation of the training data and significantly improves performance. Moreover, the NCA has the advantage of simplifying the overall training procedure, as the hyperparameters for the creation of episodes $\{w, n, m\}$ no longer need to be considered.

## 3.5 Comparison with recent methods

We now evaluate our models on three popular FSL datasets to contextualise their performance with respect to the recent literature. When considering which methods to compare against, we chose those *a)* which have been recently published, *b)* that use a ResNet-12 architecture [24] (the most commonly used), and *c)* with a setup that is not significantly more complicated than ours. For example, we only report results for the main approach of Tian et al. [44]. We omit their self-distillation [16] variant, as it can be applied to most methods and involves multiple stages of training.

Results can be found in Table 2. Besides the results for the NCA loss, we also report PNs and MNs results with both the episodic setup from Snell et al. [40] and the best one (batch size 512, 5-shot, $m + n$=16 for both PNs and MNs) found from the experiment of Fig. 2, which brings a considerable improvement over the original and other PNs implementations (See Appendix I for a comparison of our PNs implementation to other works). Note that our aim is not to improve the state of the art, but rather to shed light on the practice of episodic learning. Nonetheless, our vanilla NCA is competitive and sometimes even superior to recent methods, despite being extremely simple. It fares surprisingly well against methods that use meta-learning (and episodic learning), and also against the high-performing simple baselines based on pre-training with the cross-entropy loss. Moreover, because of the explicit inductive bias that it encodes in terms of relative position in the embedding space of samples from the same class, the NCA loss is a useful tool to consider *alongside* cross-entropy trained baselines.

## 4   Related work

Pioneered by Utgoff [46], Schmidhuber [38, 39], Bengio et al. [4] and Thrun [43], the general concept of meta-learning is several decades old (for a survey see [47, 22]). However, in the last few years it has experienced a surge in popularity, becoming the most used paradigm for learning from very few examples. Several methods addressing the FSL problem by learning on episodes were proposed. MANN [37] uses a Neural Turing Machine [21] to save and access the information useful to meta-learn; Bertinetto et al. [6] and Munkhdalai et al. [25] propose a deep network in which a "teacher" branch is tasked with predicting the parameters of a "student" branch; Matching Networks [48] and Prototypical Networks [40] are two nonparametric methods in which the contributions of different examples in the support set are weighted by either an LSTM or a softmax over the cosine distances for Matching Networks, and a simple average for Prototypical Networks; Ravi and Larochelle [32] propose instead to use an LSTM to learn the hyperparameters of SGD, while MAML [14] learns to fine-tune an entire deep network by backpropagating through SGD. Despite these works widely differing in nature, they all stress on the importance of organising training in a series of small learning problems (*episodes*) that are similar to those encountered during inference at test time.

In contrast with this trend, a handful of papers have recently shown that simple approaches that forego episodes and meta-learning can perform well on FSL benchmarks. These methods all have in common that they pre-train a feature extractor with the cross-entropy loss on the "meta-training classes" of the dataset. Then, at test time a classifier is adapted to the support set by weight imprinting [29, 11], fine-tuning [9], transductive fine-tuning [11] or logistic regression [44]. Wang et al. [49] suggest performing test-time classification by using the label of the closest centroid to the query image. Unlike these papers, which propose new methods, we are more focussed on shedding light on the possible causes behind the inefficiency of popular nonparametric few-shot learning algorithms such as Prototypical and Matching Networks.

Despite maintaining a support and a query set, the work of Raghu et al. [31] is similar in spirit to ours, and modifies episodic learning in MAML, showing that performance is almost entirely preserved when only updating the network head during meta-training and meta-testing. In this paper, we focussed on FSL algorithms just as established, and uncovered inefficiencies that not only allow for notable conceptual simplifications, but that also bring a significant boost in performance. Two related but different works are the ones of Goldblum et al. [20] and Fei et al. [12]. The former addresses PNs' poorly representative samples by training on episodic pairs with the same classes (but different instances) and using a regularizer enforcing consistency across them. The latter investigates meta-learning methods with *parametric* base learners, and shows interesting findings on the importance of having tightly clustered classes in feature space, which inspires a regularizer that improves non meta-learning models. Bai et al. [3] also show that the episodic strategy in meta-learning is inefficient by providing both theoretical and experimental arguments on methods solving a convex optimization problem at the level of the base learner. Similar to us, though via a different analysis, they show that the classic split is inefficient. Chen et al. [8] derive a generalisation bound for algorithms with a support/query separation. They do not provide any bounds for methods like NCA, which would be an interesting direction for future work. Triantafillou et al. [45] ignore the query/support separation in order to exploit all the available samples while working in a Structured SVM framework. Though the reasoning about batch exploitation is analogous to ours, the scope of the paper is very different. Finally, two recent meta-learning approaches based on Gaussian Processes [28, 41] also

merge the support and query sets during learning to take full advantage of the available data within each episode.

## 5 Conclusion

Towards the aim of understanding the reasons behind the poor competitiveness of meta-learning methods with respect to simple baselines, in this paper we investigate the role of episodes in popular nonparametric few-shot learning methods. We found that their performance is highly sensitive to the set of hyperparameters used to sample these episodes. By replacing the Prototypical Networks and Matching Networks losses with the closely related (and non-episodic) Neighbourhood Component Analysis, we were able to ignore these hyperparameters, while improving the few-shot classification accuracy. We found out that the performance discrepancy is in large part caused by the separation between support and query set within each episode, which negatively affects the number of pairwise distances contributing to the loss. Moreover, with nonparametric few-shot approaches, the episodic strategy is almost empirically equivalent to randomly discarding a portion of the distances available within a batch. Finally, we showed that our variant of the NCA achieves an accuracy on multiple popular FSL benchmarks that is competitive with recent methods, making it a simple and appealing baseline for future work.

**Broader impact.** We believe that progress in few-shot learning is important, as it can significantly impact important problems such as drug discovery and medical imaging. We also recognise that the capability of leveraging very small datasets might constitute a threat if deployed for surveillance by authoritarian entities (e.g. by applying it to problems such as re-identification and face recognition).

## Acknowledgements and Disclosure of Funding

We would like to thank Jõao F. Henriques, Nicholas Lord, John Redford, Stuart Golodetz, Sina Samangooei, and the anonymous reviewers for insightful discussions and helpful suggestions to improve the manuscript and our analysis in general. This work was supported by Five AI Limited.

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
