# SUPPLEMENTARY MATERIAL
# Episodic Learning in Nonparametric
# Few-shot Classification: A Case Study

## A   Number of pairs lost with episodic batches

In this section we demonstrate that the total number of training pairs that the NCA loss can exploit within a batch is always strictly superior or equal to the one exploited by the episodic batch strategy used by Prototypical Networks (PNs) and Matching Networks (MNs).

To ensure we have a "valid" episodic batch with a nonzero number of both positive and negative distance pairs, we assume that $n, m \geq 1$, and $w \geq 2$. Below, we show that the number of positives for the NCA, i.e. $\binom{m+n}{2}w$, is always greater or equal than the one for PNs and MNs, which is $mnw$:

$$
\begin{aligned}
\binom{m+n}{2}w &= \frac{(m+n)!}{2!(m+n-2)!}w \\
&= \frac{1}{2}(m+n)(m+n-1)w \\
&= \frac{1}{2}(m^2 + 2mn - m + n^2 - n)w \\
&= \frac{1}{2}(m(m-1) + 2mn + n(n-1))w \\
&\geq \frac{1}{2}(2mn)w = wmn.
\end{aligned}
$$

Similarly, we can show for negative distance pairs that $\binom{w}{2}(m+n)^2 > w(w-1)mn$:

$$
\begin{aligned}
\binom{w}{2}(m+n)^2 &= \frac{w!}{2!(w-2)!}(m^2 + 2mn + n^2) \\
&= \frac{1}{2}w(w-1)(m^2 + 2mn + n^2) \\
&> \frac{1}{2}w(w-1)(2mn) \\
&= w(w-1)mn.
\end{aligned}
$$

This means that the NCA has at least the same number of positives as Prototypical and Matching Networks, and has always strictly more negative distances. In general, the total number of *extra* pairs that NCA can rely on is $\frac{w}{2}(w(m^2 + n^2) - m - n)$.

35th Conference on Neural Information Processing Systems (NeurIPS 2021).

## B  Details about the ablation studies of Section 3.4

Referring to the three key differences between the losses of Prototypical Networks, Matching Networks, and the NCA listed in Sec. 2.2, in this section we detail how to obtain the ablations we used to perform the experiments of Sec. 3.4.

We can "disable" the creation of prototypes (point II), which will change the prototypical loss $\mathcal{L}_{\text{PNs}}$ to the Matching Networks loss $\mathcal{L}_{\text{MNs}}$.

$$\mathcal{L}(S,Q) = \frac{-1}{|Q|} \sum_{\substack{(\mathbf{q}_i,y) \\ \in Q}} \log \left( \frac{\sum\limits_{\substack{(\mathbf{s}_j,y') \\ \in S_y}} \exp -\|\mathbf{q}_i - \mathbf{s}_j\|^2}{\sum\limits_{\substack{(\mathbf{s}_k,y'') \\ \in S}} \exp -\|\mathbf{q}_i - \mathbf{s}_k\|^2} \right).$$

This is similar to $\mathcal{L}_{\text{NCA}}$, where the positives are represented by the distances from $Q$ to $S_k$, and the negatives by the distances from $Q$ to $S \setminus S_k$. The only difference now is the separation of the batch into a query and support set.

Independently, we can "disable" point 1 for $\mathcal{L}_{\text{PNs}}$, which gives us

$$\mathcal{L}(S,Q) = \frac{-1}{|Q|+|C|} \sum_{\substack{(\mathbf{z}_i,y_i) \\ \in Q\cup C}} \log \left( \frac{\sum\limits_{\substack{(\mathbf{z}_j,y_j) \\ \in Q\cup C \\ y_j=y_i \\ i\neq j}} \exp -\|\mathbf{z}_i-\mathbf{z}_j\|^2}{\sum\limits_{\substack{(\mathbf{z}_k,y_k) \\ \in Q\cup C \\ k\neq i}} \exp -\|\mathbf{z}_i-\mathbf{z}_k\|^2} \right),$$

which essentially combines the prototypes with the query set, and computes the NCA loss on that total set of embeddings.

Finally, we can "disable" both point 1 and 2 for both $\mathcal{L}_{\text{MNs}}$ and $\mathcal{L}_{\text{PNs}}$, which gives us

$$\mathcal{L}(S,Q) = \frac{-1}{|Q|+|S|} \sum_{\substack{(\mathbf{z}_i,y_i) \\ \in Q\cup S}} \log \left( \frac{\sum\limits_{\substack{(\mathbf{z}_j,y_j) \\ \in Q\cup S \\ y_j=y_i \\ i\neq j}} \exp -\|\mathbf{z}_i-\mathbf{z}_j\|^2}{\sum\limits_{\substack{(\mathbf{z}_k,y_k) \\ \in Q\cup S \\ k\neq i}} \exp -\|\mathbf{z}_i-\mathbf{z}_k\|^2} \right).$$

This almost exactly corresponds to the NCA loss, where the only difference is the construction of batches with a fixed number of classes and a fixed number of images per class.

## C  Differences between the NCA and contrastive losses

$\mathcal{L}_{\text{NCA}}$ is similar to the contrastive loss functions [9, 3] that are used in self-supervised learning and representation learning. The main differences are that *a)* in contrastive losses, the denominator only contains negative pairs and *b)* the inner sum in the numerator is moved outside of the logarithm in the supervised contrastive loss function from Khosla et al. [9]. We opted to work with the NCA loss because we found it performs better than the supervised constrastive loss in a few-shot learning setting. Using the supervised contrastive loss we only managed to obtain 51.05% 1-shot and 63.36% 5-shot performance on the *mini*Imagenet test set.

## D  Effectiveness of different classification strategies

In Table 1, we compare the inference methods discussed in Section 2.3 using embeddings trained with the NCA loss. It might sound surprising that the *nearest centroid* approach outperforms *soft assignments*, as the latter closely reflects NCA's training protocol. We speculate its inferior

| | *mini*ImageNet | |
| --- | --- | --- |
| method | 5-shot val | 5-shot test |
| $k$-NN | $75.82 \pm 0.11$ | $73.52 \pm 0.10$ |
| SOFT ASSIGNMENTS | $79.11 \pm 0.12$ | $77.16 \pm 0.10$ |
| NEAREST CENTROID | $\mathbf{80.61 \pm 0.12}$ | $\mathbf{78.30 \pm 0.10}$ |
| | CIFAR-FS | |
| $k$-NN | $73.46 \pm 0.12$ | $80.94 \pm 0.10$ |
| SOFT ASSIGNMENTS | $76.12 \pm 0.12$ | $83.31 \pm 0.10$ |
| NEAREST CENTROID | $\mathbf{77.80 \pm 0.12}$ | $\mathbf{85.13 \pm 0.10}$ |

Table 1: Comparison in terms of accuracy between the different classifications stategies of Section 2.3 when using models trained with the NCA loss.

performance could be caused by poor model calibration [8]: since the classes between training and evaluation are disjoint, the model is unlikely to produce calibrated probabilities. As such, within the softmax, outliers behaving as false positives can happen to highly influence the final decision, and those behaving as false negatives can end up being almost ignored (their contribution is squashed toward zero). With the nearest centroid classification approach outliers might still represent an issue, but it is likely that their effect will be less dramatic.

# E  Extended discussion

**Section 3.2 – Discussion on the number of pairs per episodic setup.**  In Table 2 we plot the number of positives and negatives (gradients contributing to the loss) for the NCA loss as well as for different episodic configurations of PNs, to see whether the difference in performance can be explained by the difference in the number of distance pairs that can be exploited in a certain batch configuration. This is often true, as within each sub-table (representing a different batch size) the ranking can almost be fully explained by the number of total pairs in the rightmost column. However, there are some exceptions to this: (i) the difference between 5-shot with m+n=16 and 5-shot with m+n=8 in (for all batch sizes), and (ii) the difference between 5-shot with m+n=32 and 1-shot with m+n=8 for batch size 128.

To understand (i), we can see that the number of positive pairs is much higher for m+n=16 than for m+n=8. Since the positive pairs constitute a less frequent (and potentially more informative) training signal, this can explain the difference. The m+n=32 variant has an even higher number of positives than m+n=16, but the loss in performance there could be explained by a drastically lower number of negatives, and by the fact that the number of ways used during training is lower.

To understand (ii), we see that the number of pairs for 5-shot with m+n=32 is higher than 1-shot with m+n=8. However, due to the small batch size of 128, the number of ways during training for 5-shot with m+n=32 is only 4, whereas for 1-shot with m+n=8 it is 16. This could explain the higher performance of 1-shot with m+n=8.

So, while indeed generally speaking the higher number of pairs the better (which is also corroborated by Figure 3 from the main paper, where moving right on the x-axis sees higher performance for both NCA and PNs), one should also consider how this interacts with the positive/negative balance and the number of classes present within a batch.

**Section 3.4 – Discussion on the number of pairs used for the ablations.**  The fact that each single ablation does not have much influence on the performance, but their combination does, could be explained by the number of distance pairs exploited by the individual ablations. To illustrate this, we use the formulas described in Table 1 from the main paper to compute the number of positives and negatives used for each ablation with a batch size of 256. Looking at the ablation results shown in Figure 4 from the main paper: for row 6 there are 480 positives, and 14,880 negatives; while for row 7 there are 576 positives and 20,170 negatives. In both cases, the number is significantly lower than the corresponding NCA with a fixed batch composition, which totals 896 positives and 31,744 negatives, which could explain the gap between row 6 (and 7) and row 3. Moreover, from row 5 (and 6) to row 7 we see a modest increase in performance, which can also be explained by the slightly larger number of distance pairs.

| Batch size | 1-shot | 5-shot | Method | # pos | # neg | # tot |
|---|---|---|---|---|---|---|
| | $62.84 \pm 0.13$ | $77.07 \pm 0.10$ | *NCA w/ nearest centroid* | 1792 | 129024 | 130816 |
| | $60.53 \pm 0.13$ | $75.79 \pm 0.11$ | PNs *5-shot m+n=16* | 1760 | 54560 | 56320 |
| | $59.85 \pm 0.13$ | $74.99 \pm 0.11$ | PNs *5-shot m+n=8* | 960 | 60480 | 61440 |
| 512 | $59.82 \pm 0.13$ | $74.44 \pm 0.11$ | PNs *5-shot m+n=32* | 2160 | 32400 | 34560 |
| | $58.01 \pm 0.14$ | $71.80 \pm 0.11$ | PNs *1-shot m+n=8* | 448 | 28224 | 28672 |
| | $51.75 \pm 0.13$ | $65.77 \pm 0.12$ | PNs *1-shot m+n=16* | 465 | 14415 | 14880 |
| | $41.49 \pm 0.12$ | $52.90 \pm 0.12$ | PNs *1-shot m+n=32* | 496 | 7440 | 7936 |
| | $62.07 \pm 0.14$ | $76.26 \pm 0.10$ | *NCA w/ nearest centroid* | 384 | 32256 | 32640 |
| | $59.60 \pm 0.13$ | $74.64 \pm 0.11$ | PNs *5-shot m+n=16* | 880 | 13200 | 14080 |
| | $58.80 \pm 0.13$ | $74.03 \pm 0.11$ | PNs *5-shot m+n=8* | 480 | 14880 | 15360 |
| 256 | $57.62 \pm 0.13$ | $72.53 \pm 0.11$ | PNs *5-shot m+n=32* | 1080 | 7560 | 8640 |
| | $57.61 \pm 0.14$ | $71.49 \pm 0.11$ | PNs *1-shot m+n=8* | 224 | 6944 | 7168 |
| | $48.46 \pm 0.13$ | $61.64 \pm 0.12$ | PNs *1-shot m+n=16* | 240 | 3600 | 3840 |
| | $37.96 \pm 0.11$ | $48.38 \pm 0.11$ | PNs *1-shot m+n=32* | 248 | 1736 | 1984 |
| | $58.36 \pm 0.14$ | $72.69 \pm 0.11$ | *NCA w/ nearest centroid* | 64 | 8064 | 8128 |
| | $57.02 \pm 0.13$ | $72.05 \pm 0.11$ | PNs *5-shot m+n=16* | 440 | 3280 | 3520 |
| | $56.45 \pm 0.13$ | $71.42 \pm 0.11$ | PNs *5-shot m+n=8* | 240 | 3600 | 3840 |
| 128 | $49.88 \pm 0.13$ | $64.50 \pm 0.11$ | PNs *5-shot m+n=32* | 540 | 1620 | 2160 |
| | $51.75 \pm 0.13$ | $65.77 \pm 0.12$ | PNs *1-shot m+n=8* | 112 | 1680 | 1792 |
| | $40.46 \pm 0.12$ | $52.27 \pm 0.11$ | PNs *1-shot m+n=16* | 120 | 840 | 960 |
| | $28.54 \pm 0.09$ | $33.50 \pm 0.09$ | PNs *1-shot m+n=32* | 124 | 372 | 496 |

Table 2: Number of positives and negatives used in the experiments of Figure 2 from the main paper. We also report the 1-shot and 5-shot accuracies on the validation set of CIFAR-FS using PNs and NCA with nearest centroid classification.

## F   Implementation details

**Benchmarks.**   In our experiments, we use three popular FSL benchmarks. *mini*ImageNet [15] is a subset of ImageNet generated by randomly sampling 100 classes, each with 600 randomly sampled images. We adopt the commonly used splits from [12], with 64 classes for meta-training, 16 for meta-validation and 20 for meta-testing. **CIFAR-FS** [1] is an anagolous version of *mini*Imagenet, but for CIFAR-100. It uses the same sized splits and same number of images per split. *tiered*ImageNet [13] is also constructed from ImageNet, but contains 608 classes, with 351 training classes, 97 validation classes and 160 test classes. The class splits have been generated using WordNet [11] to ensure that the training classes are semantically "distant" to the validation and test classes. For all datasets, we use images of size $84 \times 84$.

**Architecture.**   In all our experiments, $f_\theta$ is represented by a ResNet12 with widths $[64, 160, 320, 640]$. We chose this architecture, initially adopted by Lee et al. [10], as it is the one which is most frequently adopted by recent FSL methods. Unlike most methods, we do not use a DropBlock regulariser [5], as we did not notice it to meaningfully contribute to performance.

**Optimisation.**   To train all the models used for our experiments, unless differently specified, we used a SGD optimiser with Nesterov momentum, weight decay of 0.0005 and initial learning rate of 0.1. For *mini*ImageNet and CIFAR-FS we decrease the learning rate by a factor of 10 after 70% of epochs have been trained, and train for a total of 120 epochs. As data augmentations, we use random horizontal flipping and centre cropping. For Section 3.5 only, we slightly change our training setup. On CIFAR-FS, we increase the number of training epochs from 120 to 240, which improved accuracy by about 0.5%. For *tiered*ImageNet, we train for 150 epochs and decrease the learning rate by a factor of 10 after 50% and 75% of the training progress. For *tiered*ImageNet only we increased the batch size to 1024, and train on 64 classes (like *mini*ImageNet and CIFAR-FS) and 16 images per class within a batch, as we found it being beneficial. These changes affect and are beneficial for all our implemented methods and baselines: NCA, Prototypical Networks, and Matching Networks (with both old and new batch setup), and SimpleShot [16].

**Projection network.**   Similarly to [9, 3], we also experimented (for MNs, PNs, and NCA) with a *projection network* (but *only* for the comparison of Section 3.5). The projection network is a single linear layer $A \in \mathbb{R}^{M \times P}$ that is placed on top of $f_\theta$ at training time, where $M$ is the output dimension of the neural network $f_\theta$ and $P$ is the output dimension of $A$. The output of $A$ is only used during

training. At test time, we do not use the output of $A$ and directly use the output of $f_\theta$. For CIFAR-FS and *tiered*ImageNet, we found this did not help performance. For *mini*ImageNet, however, we found that this improved performance for the NCA – but not for PNs and MNs – and we set $P = 128$. Note that this is not an unfair advantage over other methods. Compared to SimpleShot [16] and other simple baselines, given that they use an extra fully connected layer to minimise cross entropy during pre-training, we actually use fewer parameters without the projection network (effectively making our ResNet12 a ResNet11).

**Choice of hyperparameters.** During the experimental design, we wanted to ensure a fair comparison between the NCA, PNs, and MNs. As a testimony of this effort, we obtained very competitive results for PNs (see for example the comparison to recent papers where architectures of similar capacity were used [16, 4]). In particular:

- We always use the normalisation strategy of SimpleShot [16], as it is beneficial also for both PNs and MNs.
- Unless expressively specified, we always used PNs' and MNs' 5-shot model, which in our implementation outperforms the 1-shot model (for both 1-shot and 5-shot evaluation). Instead, [14, 15] train and tests with the same number of shots.
- Apart from the episodes hyperparameters of PNs and MNs, which we did search and optimise over to create the plots of Figure 3.2 (main paper), the only other hyperparameters of PNs and MNs are those related to the training schedule, which are the same as the NCA. To set them, we started from the simple SGD schedule used by SimpleShot [16] and only marginally modified it by increasing the number of training epochs to 120, increasing the batch size to 512 and setting weight decay and learning rate to $5e - 4$ and $0.1$, respectively. To decide this setting, we run a small grid search and empirically verified that it is the best for all three methods. Moreover, as a sanity check we trained both the NCA and PNs with the exact training schedule used by SimpleShot [16]. Results are reported in Table 3, and show that the schedule we used for this paper is considerably better for both PNs and NCA. In general, we observed that the modifications were beneficial for NCA, PNs and MNs, and improvements in performance in NCA, PNs and MNs were highly correlated. This is to be expected given the high similarity between the three methods and losses.

Computing infrastructure details. For our experiments we had eight NVIDIA GeForce GTX 1080 Ti GPUs available. However, for each model we usually only needed 2 GPUs to train, except for the *tiered*ImageNet experiments using a batch size of 1024 where we needed 4 GPUs. All our *mini*ImageNet and CIFAR-FS experiments took about 2 hours to finish training, and the *tiered*ImageNet experiments took 8 hours per model.

# G   Performance improvements

**Adapting to the support set.** None of the algorithms we considered perform any kind of parameter adaptation at test time. On the one hand this is convenient, as it allows fast inference; on the other hand, useful information from the support set $S$ might remain unexploited.

In the 5-shot case it is possible to minimise the NCA loss since it can directly be computed on the support set: $\mathcal{L}_{\text{NCA}}(S)$. We tried training a positive semi-definite matrix $A$ on the outputs of the trained neural network, which corresponds to learning a Mahalanobis distance metric as in Goldberger et al. [7]. However, we found that there was no meaningful increase in performance. Differently, we did find that fine-tuning the whole neural network $f_\theta$ by $\arg\min_\theta \mathcal{L}_{\text{NCA}}(S)$ is beneficial (see Table 4). Nonetheless, given the computational cost, we opted for non performing adaptation to the support sets in any of our experiments.

**Features concatenation.** For NCA, we also found that concatenating the output of intermediate layers modestly improves performance at (almost) no additional cost. We used the output of the average pool layers from all ResNet blocks except the first and we refer to this variant as NCA multi-layer. However, since this is an orthogonal variation, we do not consider it in any of our experiments. Results on *mini*ImageNet and CIFAR-FS are shown in Table 4.

|  | *mini*ImageNet | | CIFAR-FS | |
| --- | --- | --- | --- | --- |
| **method** | **1-shot** | **5-shot** | **1-shot** | **5-shot** |
| **PNs (SimpleShot)** | $57.99 \pm 0.21$ | $74.33 \pm 0.16$ | $53.76 \pm 0.22$ | $68.54 \pm 0.19$ |
| **PNs (ours)** | $62.79 \pm 0.12$ | $78.82 \pm 0.09$ | $59.60 \pm 0.13$ | $74.64 \pm 0.11$ |
| **NCA (SimpleShot)** | $61.21 \pm 0.22$ | $76.39 \pm 0.16$ | $59.41 \pm 0.24$ | $73.29 \pm 0.19$ |
| **NCA (ours)** | $64.94 \pm 0.13$ | $80.12 \pm 0.09$ | $62.07 \pm 0.14$ | $76.26 \pm 0.10$ |

Table 3: Comparison (on the validation set of *mini*ImageNet and CIFAR-FS) between using the hyper-parameters from SimpleShot [16] and the ones from this paper (**ours**). Models have been trained with batches of size 256, as in [16]. PNs episodic batch setup uses $m + n$=16, as it is the highest performing one. NCA is evaluated using nearest centroid classification.

|  | *mini*ImageNet | | CIFAR-FS | |
| --- | --- | --- | --- | --- |
| **method** | **1-shot** | **5-shot** | **1-shot** | **5-shot** |
| **NCA** | $62.52 \pm 0.24$ | $78.3 \pm 0.14$ | $72.48 \pm 0.40$ | $85.13 \pm 0.29$ |
| **NCA multi-layer** | $63.21 \pm 0.08$ | $79.27 \pm 0.08$ | $72.44 \pm 0.36$ | $85.42 \pm 0.29$ |
| **NCA (ours) multi-layer + ss** | - | $\mathbf{79.79 \pm 0.08}$ | - | $\mathbf{85.66 \pm 0.32}$ |

Table 4: Comparison between vanilla NCA, NCA using multiple evaluation layers and NCA performing optimisation on the support set (ss). The NCA can only be optimised in the 5-shot case, since there are not enough positives distances in the 1-shot case. Optimisation is conducted for 5 epochs using Adam, with learning rate 0.0001 and weight decay 0.0005. NCA is always evaluated using nearest centroid classification.

# H   Additional results for Section 3.2

Figure 1 and Figure 2 complement the results of Figure 2 and Figure 4 from the main paper, respectively. As can be seen, these figures support the main conclusions made in the paper: in Figure 1 we can see that the NCA loss outperforms all the episodic setups also on *mini*ImageNet. Note that, since there are no prototypes in Matching Networks, Figure 2 only has one ablation. It is then clear from the figure that, similarly to what we showed for Prototypical Networks in Figure 4 from the main paper, disregarding the separation between support and query set is beneficial for Matching Networks as well.

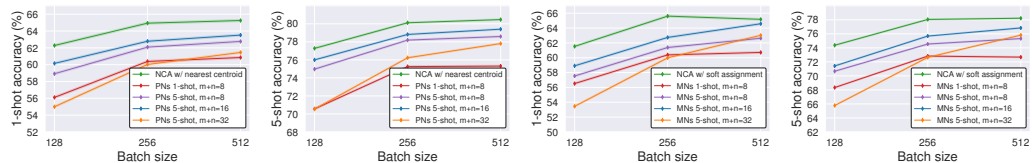

Figure 1: 1-shot and 5-shot accuracies on *mini*ImageNet (val. set) for Prototypical and Matching Networks models trained with different episodic configurations: 1-shot with $m + n$=8 and 5-shot with $m + n$=8, 16 or 32. NCA models are trained on batches of size 128, 256 and 512 to match the size of the episodes. Reported values correspond to the mean accuracy of three models trained with different random seeds and shaded areas represent 95% confidence intervals. See Sec. 3.2 for details.

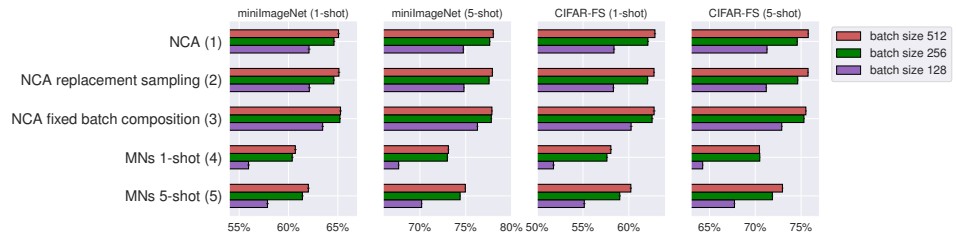

Figure 2: Ablation experiments on NCA and Matching Networks, both on batches (or episodes) of size 128, 256, and 512 on *mini*ImageNet and CIFAR-FS (val. set). Reported values correspond to the mean accuracy of three models trained with different random seeds and error bars represent 95% confidence intervals. See Sec. 3.4 for details.

# I   Comparing to other PN implementations

In Table 5 we compare our implementation and hyperparameter selection for PNs to previous works that have re-implemented PNs using variants of ResNet. We can see that our implementation achieves the best results, indicating that the better performance achieved by NCA is not the result of an uneven hyperparameter search.

| PNs Implementation | Architecture | miniImageNet | |
| --- | --- | --- | --- |
| | | **1-shot** | **5-shot** |
| [2] | ResNet10 | $51.98 \pm 0.84$ | $72.64 \pm 0.64$ |
| [2] | ResNet18 | $54.16 \pm 0.82$ | $73.68 \pm 0.65$ |
| [2] | ResNet34 | $53.90 \pm 0.83$ | $74.65 \pm 0.64$ |
| [6] | WideResNet-28-10 | $55.85 \pm 0.48$ | $68.72 \pm 0.36$ |
| [10] | ResNet12 | $59.25 \pm 0.64$ | $75.60 \pm 0.84$ |
| Ours ([14] episodes) | ResNet12 | $59.78 \pm 0.12$ | $75.42 \pm 0.09$ |
| Ours (best episodes) | ResNet12 | $\mathbf{61.32 \pm 0.12}$ | $\mathbf{77.77 \pm 0.09}$ |

Table 5: Comparison (on the test set *mini*ImageNet) across papers between implementations of PNs [14] on ResNet architectures. Our best episode configuration is selected from experiments in Section 3.2.