# OpenReview forum: "On Episodes, Prototypical Networks, and Few-Shot Learning"
_NeurIPS.cc/2021/Conference — NeurIPS 2021 Poster_

### Official Review · Reviewer_1YGQ · 2021-07-01

**Rating:** 6
**Confidence:** 4

**Summary:**

This paper presents a new perspective on the episodic learning of nonparametric few-shot learning methods. The main claim is that current popular nonparametric methods, such as Prototypical Networks (PNs) and Matching Networks (MNs), are not data-efficient because less gradient signal is propagated during training due to the artificial division of the data points to support and query sets. The authors instead propose to use the standard learning protocol with batches and an equivalent loss function based on the Neighbourhood Component Analysis (NCA). This loss function exploits all connections between data points in the batch. The authors then propose three techniques to perform a few-shot classification during evaluation based on k-NN, nearest centroid, and soft assignment. Through extensive experiments, the authors justify their perspective and show comparable results to other recent FSL methods (not necessarily nonparametric ones).

**Limitations And Societal Impact:**

The authors addressed the potential limitations and negative societal impact of their work.

**Main Review:**

Overall it is an interesting paper, and I think it should be accepted.

Strengths:
- The claim that the hyperparameters controlling the episode creation can significantly affect the performance makes a lot of sense.
- The paper presents extensive empirical justifications for the main claims presented in it.
- Despite the method's simplicity it shows good results on relevant benchmarks.
- The paper is written clearly and easy to understand.
- The results in the paper are easily reproducible. The code is very organized and full experimental details were given for all experiments.

Some comments:
- Although NCA is compared against relatively recent baselines and it is not delivered as a new SoTA, it seems that there is a noticeable gap between its performance and more recent baselines (e.g., [1, 2]). Furthermore, this paper scope is limited only to nonparametric metric-based learners. Therefore, I think the authors should add a discussion on how to extend their approach beyond the current scope and how it can be combined with more recent approaches.
- I think that the authors should have addressed other types of non-parametric few-shot learners. For example, recently two approaches based on Gaussian processes were proposed [3, 4]. Although these methods use the standard episodic learning, they ignore the common support/query split (as was done in the ablation study in this paper).
- In Table 2 -> CIFAR-FS -> 5-shot, RFIC-simple achieves the highest accuracy and therefore should be in bold.

General wonderings:
- Line 175 states "Each model is evaluated on 10,000 episodes sampled from the validation or the test set". Does this mean that you didn't use the pre-defined validation set for hyper-parameter selection and early stopping? If so, how did you perform these operations?  Is it consistent with the results reported for the baseline methods in Table 2?
- When comparing Fig. 3 in the main text with Fig. 1 in the Appendix it seems that using 80% of the pairs yield similar results to those obtained with all the pairs. Is it true? If so, how can you justify this?
- Following the previous point, did you try other (more advanced) techniques for pair selection? As a motivation, it may allow achieving similar accuracy while reducing the computational burden required by the denominator of the NCM loss.


[1] Rizve, M. N., Khan, S., Khan, F. S., & Shah, M. (2021). Exploring Complementary Strengths of Invariant and Equivariant Representations for Few-Shot Learning. In Proceedings of the IEEE/CVF Conference on Computer Vision and Pattern Recognition (pp. 10836-10846).
[2] Zhen, X., Du, Y., Xiong, H., Qiu, Q., Snoek, C. G., & Shao, L. (2020). Learning to learn variational semantic memory. arXiv preprint arXiv:2010.10341.
[3] Patacchiola, M., Turner, J., Crowley, E. J., O'Boyle, M., & Storkey, A. J. (2020). Bayesian Meta-Learning for the Few-Shot Setting via Deep Kernels. Advances in Neural Information Processing Systems, 33.
[4] Snell, J., & Zemel, R. (2020, September). Bayesian Few-Shot Classification with One-vs-Each Pólya-Gamma Augmented Gaussian Processes. In International Conference on Learning Representations.

**Time Spent Reviewing:**

10

---

> ### Author Response · Authors · 2021-08-10
> **Response to reviewer 1YGQ**
>
> We thank reviewer 1YGQ for their time and constructive feedback. Please note that, for convenience, in this comment we refer to papers with letters. The list of references is given at the end of the answer.
>
>
> ----
> **1.**
> > _”Although NCA is compared against relatively recent baselines and it is not delivered as a new SoTA, it seems that there is a noticeable gap between its performance and more recent baselines (e.g., [A, B]).”_
>
> Thanks for the references, which we will add to the paper. However, we do not believe that the methods of [A] and [B] can be considered as _baselines_ in the same way the NCA, SimpleShot [C] or RFIC [D] are. More specifically:
> [A] proposes to extend the cross-entropy-based approach from [D] to enforce invariance and equivariance to a general set of geometric transformations. They do so by organising training in a multi-task fashion with three extra “heads”: one supervised and two self-supervised (which present a limited amount of extra parameters). The observations made by the authors to motivate these changes are valid for most baselines (including the NCA), and are likely to improve our results as well.
> By reading the paper, it looks like RFIC (against which we are competitive) is the baseline of [A]; as a matter of fact, that’s the performance indicated in Figure 1 of [A] as “Baseline” (the performance in the barchart is the one of RFIC in Table 3 of [A]).
>
> A similar consideration can be done for [B], whose baseline is actually Prototypical Networks (see their equation 1). In this case, the contribution is to extend the very limited “short-term” memory represented by the support set of a new test episode by introducing an _external memory module_ to store long-term semantic information acquired during training.
> We argue that in this case too the complexity of the method with respect to simple approaches like NCA, SimpleShot or RFIC is a) non-trivial and b) it could be extended to many methods.
> For [B], the most reasonable baseline would be Prototypical Networks. However, unfortunately we did not find the results of their baseline implementation for a deep embedding in the experimental results.
>
> ----
> **2.**
>
> > _“I think the authors should add a discussion on how to extend their approach beyond the current scope and how it can be combined with more recent approaches.”_
>
> Thanks for the suggestion - we will make sure to discuss this better in the paper.
> To clarify our choices: In our paper we decided to focus on Matching and Prototypical Networks to have a “clean” and unambiguous experimental evaluation. We believe that, given the evidence provided, similar conclusions to the ones drawn from our paper should apply to metric-learning-based methods in which the separation between support and query set is cosmetic rather than functional. This should especially apply to methods that are directly built on Prototypical Networks such as [B], [E], [F] and [G]. However, it is important to mention a caveat here.Though there is strong evidence suggesting this, it should be verified empirically case by case by thoroughly conducting a similar set of experiments with more recent methods after having integrated them in the same codebase.
> As mentioned in the paper, we opted for inspecting the simplest possible variant of metric-based episodic approaches to have a clean experimental setup and avoid having possible confounding factors to disentangle in our experimental analysis.
>
> ----
> **3.**
>
> > _“RFIC-simple achieves the highest accuracy [in CIFAR-FS 5-shots] and therefore should be in bold.”_
>
>
> Whops - apologies, we must have added RFIC in a second moment and forgot to fully update the  bolding. Thanks for pointing this out, we will fix it.
>
> ----
> **4.**
> > _“Did you try other (more advanced) techniques for pair selection?”_
>
>
> To keep the focus on the significance of episodes, in this work we opted for using the simplest sampling strategy for batch selection we could think of, i.e. random sampling. However, we agree it is an promising/important avenue for future work.
> For example, one could take inspiration from the paper “Sampling Matter in Deep Embedding Learning” [H], as similar conclusions should apply to the few-shot learning scenario.
> In particular, their strategy draws samples uniformly according to their relative distance. This significantly stabilises training by diminishing the variance of gradients, and turns out to be much more impactful than the specific choice of loss function.
>
>
>
> ----
> **5.**
> > _“Two approaches based on Gaussian processes were proposed [I, J]. Although these methods use the standard episodic learning, they ignore the common support/query split (as was done in the ablation study in this paper).”_
>
> Thanks for the relevant references - we will include them in the paper. Note that, while they do ignore the support/query separation, they do not particularly comment on it nor perform ablative experiments. For instance, [J] simply mentions _“As suggested by [I] we merge the support and query sets during learning to take full advantage of the available data within each episode.”_.
> Hence, it should be reasonable to assume that both [I] and [J] reached similar conclusions to ours for their Gaussian Process based approaches.
>
>
> ----
> **6.**
> > _“Line 175 states ‘Each model is evaluated on 10,000 episodes sampled from the validation or the test set’. Does this mean that you didn't use the pre-defined validation set for hyper-parameter selection and early stopping?”_
>
> Apologies for the unclear sentence, which we will edit. We simply meant that during both validation and testing we sampled 10k episodes (30k if we consider the multiple seeds used) from their respective splits. We did use the standard splits and our setup is consistent across all the methods we compare against.
>
>
> ----
> **7.**
> > _“When comparing Fig. 3 in the main text with Fig. 1 in the Appendix it seems that using 80% of the pairs yields similar results to those obtained with all the pairs. Is it true? If so, how can you justify this?”_
>
> They are indeed very close. This can also be guessed by extrapolating the trend from Figure 3,which shows how the doubling of pairs while moving on the x-axis provides diminishing returns. Given this trend, passing from 80% to 100% is fairly negligible, which is reflected in the marginal performance difference. We believe that this is caused by the fact that while increasing the density of sampling it becomes more and more likely to contribute to the loss with redundant training signal. Note that this doesn’t justify the use of the support/query separation. However, it does point to the direction of more advanced sampling strategies such as the one discussed in answer 4 above.
>
> ----
> **References**
> * [A] Exploring Complementary Strengths of Invariant and Equivariant Representations for Few-Shot Learning; Rizve et al.; CVPR 2021
> * [B] Learning to Learn Variational Semantic Memory; Zhen et al.; NeurIPS 2020
> * [C] Rethinking Few-shot Image Classification: A Good Embedding is All You Need?; Tian et al.; arXiv 2020
> * [D] SimpleShot: Revisiting Nearest-Neighbor Classification for Few-Shot Learning; Wang et al.; arXiv 2019
> * [E] Few-Shot Learning via Embedding Adaptation with Set-to-Set Functions; Ye et al.; CVPR 2020
> * [F] TapNet: Neural Network Augmented with Task-Adaptive Projection for Few-Shot Learning; Yoon et al.; ICML 2019
> * [G] TADAM: Task dependent adaptive metric for improved few-shot learning; Oreskin et al.; NeurIPS 2018
> * [H] Sampling Matters in Deep Embedding Learning; Wu et al.; ICCV 2017
> * [I] Bayesian Meta-Learning for the Few-Shot Setting via Deep Kernels; Patacchiola et al.; NeurIPS 2020
> * [J] Bayesian Few-Shot Classification with One-vs-Each Pólya-Gamma Augmented Gaussian Processes; Snell & Zemel; ICLR 2021

---

> > ### Comment · Reviewer_1YGQ · 2021-08-30
> > **Response to the Rebuttal**
> >
> > I would like to thank the authors for the detailed response. I acknowledge that the main goal of this paper is to understand the inefficiency of some of the most popular episodic methods; however, I still believe that the scope of this paper is rather limited. Having said that, I think that this paper presents interesting claims, which generally seem valid, and touches upon important and common FSL methods. Therefore, I believe it is a relevant paper for the community and keep my score intact.

---

### Official Review · Reviewer_XK2J · 2021-07-14

**Rating:** 5
**Confidence:** 5

**Summary:**

This paper conducts a case study for the non-parametric few-shot classification methods (e.g. Prototypical Networks).
It proposes to utilize the classic Neighbourhood Component Analysis (NCA) sampling instead of the original matching or prototypical style episode sampling.
The authors conducted ablation experiments to investigate the properties of this new sampling and compare it with the basic Prototypical Networks (PNs) method.
The final accuracy is comparable with the recent methods on three benchmark datasets.

**Limitations And Societal Impact:**

The main limitation is the lack of comparison and analysis with related work that also studies the effect of episode training.
In Table 2, there is a typo "MetaOptNet-SVM[21]".


**Main Review:**

Recently, there are several papers focusing on whether meta-learning or episode training is beneficial in few-shot classification, e.g [8,17,27,38]. These papers discuss the necessity of meta-learning and some alternatives such as regularization, pre-training, feature reuse. In this situation, the new findings and overall novelty of this paper are limited.

Moreover, in both the technical and experimental parts, there is a lack of thorough comparison with existing studies on the episode or meta-learning mechanisms.
Most of the ablation studies are against the PNs method, which is relatively poor. In Table 2, some related studies such as [17] are missing.

in [38], the authors proposed a simple pre-training + classifier learning baseline. The proposed NCA method seems to underperform this simple method, which weakens the significance of proposing such a new sampling strategy.

In Table 1, an extra number of gradients are required by NCA. It is unclear if this method will be less efficient than PNs.

------------------------------------------------------------------------------------------------------------------------------------------------------------------------
After Rebuttal.

After the rebuttal and discussion with other reviewers, I would like to increase my score to 5. But I still have some concerns about the limited scope and lack of comparisons.

**Time Spent Reviewing:**

1 hour

---

> ### Author Response · Authors · 2021-08-10
> **Response to reviewer XK2J**
>
> We thank reviewer XK2J for their time and feedback.
> Please note that, for convenience, in this comment we refer to papers using the list of references from the main paper.
>
> ----
> **1.**
> > _“Recently, there are several papers focusing on whether meta-learning or episode training is beneficial in few-shot classification, e.g [8,17,27,38]. [...] the new findings and overall novelty of this paper are limited.”_
>
> > _“The main limitation is the lack of comparison and analysis with related work that also studies the effect of episode training.”_
>
> Broadly speaking, our paper and the ones mentioned (plus several others cited in our submission) all have the general theme of “tackling few-shot classification without meta-learning”.
> However, we disagree that our work’s novelty is limited: these papers and ours complement each other.
> Papers such as [8] and [38] present simple FSL methods without meta-learning, but they only introduce new competitive baselines, not an analysis that aims at understanding the inefficiencies of the episodic strategy. Their results (together with others such as [7,9,42]) actually inspired us to conduct an in-depth investigation.
>
> [17] and [27] are more similar in spirit to our work, as we mentioned in the last paragraph of the related work. [27] presents a thorough analysis of MAML, which shows that it is only useful to update a small part of the network’s parameters (the head). [17] analyses meta-learning approaches which solve a convex-optimisation problem at the level of the base learner ([4] and [21]) and observe how the features these methods learn “tend to cluster object classes more tightly in feature space” and propose regularizers that induce this property.
>
> Our work is significantly different from prior work: it presents an in-depth analysis of two very popular non-parametric episodic methods that illustrates the reasons behind their inefficacy. As a byproduct of this analysis, we also present another simple and competitive baseline that can be considered alongside the ones already introduced by work like [7,8,42,9,38]. However, we would like to stress on the fact that our main contribution is the _analysis_, not the baseline.
>
> To summarise: As mentioned in the answer above, we believe our work is novel complementary to existing work, which we did our best to reference in our paper. Please let us know if there are studies on episodic learning that we are still missing, and we will be happy to comment on them in the discussion and add them to our paper.
>
> ----
> **2.**
> > _“in [38], the authors proposed a simple pre-training + classifier learning baseline. The proposed NCA method seems to underperform this simple method, which weakens the significance of proposing such a new sampling strategy.”_
>
> As we mentioned in the text (line 292), our work is an analysis that aims at shedding more light on episodic learning rather than improving over the state-of-the-art. Thus, Section 3.5 and Table 2 have the purpose to contextualise the performance with respect to other simple baselines rather than claiming that our version of the NCA should be adopted as it represents a new state of the art, or the best baseline around.
>
> Moreover, we would like to point out that the simple NCA actually (modestly) outperforms RFIC [38] in the one-shot setups of miniImageNet and CIFAR-FS, while it is (modestly) outperformed by RFIC for the five-shot setups for these two datasets and in both setups for tieredImageNet (of about 1%). (Apologies, we realised that the bolding of CIFAR-FS 5-shot column is incorrect, and we will fix that).
> Hence, we believe it is fair to say that these two baselines are roughly in the same league.
>
>
> ----
> **3.**
>
> > _“In Table 2, some related studies such as [17] are missing.”_
>
> As mentioned at line 283, given that our contribution is not to improve over the state of the art, we decided to only include simple methods in our comparison. We opted for not reporting [17] in the table because we believed that the regularisers introduced (to modify [21], which we report) could be used by most methods. Moreover, note that the regularisers decrease the performance of [12] in one of the two datasets reported.
> Nonetheless, we are happy to discuss this further and add these results to our table if this reviewer prefers so.
>
> ----
> **4.**
>
> > _“In Table 1, an extra number of gradients are required by NCA. It is unclear if this method will be less efficient than PNs.”_
>
> In our experiments, we did not notice a meaningful slowdown of the methods exploiting a larger number of gradients. This is likely due to the fact that in modern frameworks like PyTorch it is possible to compute similarities between all elements of a batch very efficiently.
> However, if one had to use the NCA with very large batches the difference might become significant. We discuss this in Section 3.3, where we show that in case computation would become a concern one can simply use the NCA approach while randomly discarding elements from each batch, as the added complexity of introducing hyperparameters defining the episodes offers no advantages over this simpler strategy (Figure 3).
>
> ----
> We hope that our answers satisfy the main concerns expressed. If not, we are happy to engage in further discussion.

---

### Official Review · Reviewer_GQUE · 2021-07-16

**Rating:** 5
**Confidence:** 5

**Summary:**

This paper studies the role of the popular episodic training paradigm, in the context of two metric learning-based episodic models: Prototypical Networks (PN) and Matching Networks (MN). They show that these popular methods underperform compared to the closely-related NCA model which is non-episodic, i.e. does not separate the examples sampled in each training batch into disjoint support and query sets. They argue that the superior performance of NCA is because, due to not performing that separation, the total number of pairwise comparisons that are used in the loss computation is larger than those used for PN/MN, making the gradients more informative. Indeed, in episodic models, each query example is only compared to support examples, but not to other query examples, resulting in significantly fewer comparisons. Experimentally, they show that for a fixed batch size (where the ‘batch size’ in the case of the episodes is given by the combined size of the support and query sets), NCA outperforms PN and MN, despite its simplicity in terms of its smaller number of hyperparameters. They also show that randomly discarding comparisons from NCA leads to similar performance to the analogous PN/MN models and perform a set of ablations to “bridge” the gap between PN and NCA, further strengthening their finding that support/query separation hurts performance.


**Limitations And Societal Impact:**

Yes, I believe they have addressed this adequately.

**Main Review:**

High-level review
-----------------------
The paper is really well-written and an enjoyable read. Despite the findings not being very surprising to me, the inefficiency with which episodic models exploit their training data compared to non-episodic baselines has not been formally established, and it’s nice to have the empirical evidence shown in the paper to support that intuition. It is also a nice contribution to show that the NCA model can perform well for few-shot classification, and can serve as a strong baseline alongside the simple cross-entropy classifier baselines that are gaining traction. However, the study of this work is limited to only a couple settings for the shot and way (1-shot 5-way and 5-shot 5-way), to simple benchmarks, and only two episodic methods. It is not clear if the same conclusions generalize to other models and to harder settings, e.g. varying shots at test time, and harder generalization problems to classes that are dissimilar to the training set (e.g. coming from different datasets). These aspects limit the impact of the paper in my opinion and make it hard to see what the take-away is. How should few-shot learning researchers change their approach based on this finding? PN/MN are no longer state-of-the-art models, and recent work these days is moving away from (exclusively, at least) relying on episodic training for few-shot learning (e.g. [1,2,3,4,5,6,7,8] to name a few). In some of these works, episodic training is still used for certain components of the network (e.g hypernets that learn to condition a shared network for each task by producing FiLM parameters for instance, which are naturally defined in an episodic manner), or used to fine-tune a pre-trained network, but the representation is pre-trained for a non-episodic objective. For the above reasons, I'm slightly leaning towards rejection of this paper at this stage. Please see below for more detailed comments.

Detailed comments
--------------------------
- Based on the observation that PN/MN perform significantly fewer comparisons than NCA, they underperform NCA in the settings considered and the authors conclude that NCA is preferable in those settings. However, one could argue instead that, due to this observation, the episodic models may require a significantly larger batch size to reach their potential (to increase their number of comparisons that contribute to each update), and so perhaps their poor performance is due to that hyperparameter (the batch size) selected inappropriately for these methods. Of course very large batch sizes can be challenging from an implementation standpoint (e.g. due to memory issues etc) but, to reliably explore the potential of these methods, these challenges can be overcome by carefully parallelizing across GPUs for example (or training with a meta-batch of episodes which is actually common practice). A notable example from the recent literature that yields very impressive performance but requires very large batches to do so is SimCLR [9] (it was trained with a batch size >4K). Perhaps “scaling up” episodic methods in this fashion will result in surpassing the (analogously scaled up) NCA method, so in my view the jury is still out on this comparison.

- This paper only focuses on the disadvantages of episodic models and advantages of NCA. It would also be useful to discuss episodic models’ merits. A potential advantage of PN, for example, is that we can modify its training (specifically, through the choice of the training shot hyperparameter), to specialize it to the shot setting of interest. [10] formally shows that the choice of the ‘shot’ hyperparameter during training impacts the properties of the learned embedding space with smaller shots during training focusing on minimizing intra-class variance (pulling examples of the same class together) while larger training shots on maximizing inter-class variance (pushing different classes apart). As a consequence, models with smaller training shots are more appropriate for small-shot test-time tasks, and analogously for larger-shot models. NCA on the other hand does not (explicitly at least) offer that “knob” to tune at training time in order to control the desired properties of the resulting embedding space. Can we theoretically or empirically study whether NCA-learned models are capable of performing well on tasks of a wide range of shots at test time, when using the nearest-centroid classification as the test-time algorithm? The paper only considers 1-shot and 5-shot evaluation settings which are both very small shots.

- An interesting data point to include in the discussion from the gradient-based meta-learning world is that, while MAML [11] uses a support and query separation, its first-order Reptile variant [12] does not. Is there a high-level take-away that connects the relationship of Reptile to MAML and that of PN (or MN) to NCA?

- Related work: [13] should be included in the related work in the context of few-shot learning models that do not require a support and query separation.

- Recently, there have been some theoretical papers that investigate the role of the support/query framework from different perspectives [14,15]. It would be useful to include these in the discussion and consolidate the empirical observations made here with the theoretical findings from those papers.

- In the ablations, what is the PN model where "prototypes are not computed"? PN requires prototypes, so omitting their computation is not an ablation of this model; it is a different model. Is this exactly the same as MN in this case? If so, I would just refer to it as MN.

- In the original PN paper they showed empirically that it was beneficial to "match" the shot used at training and test times. The results here contradict this, since PN 5-shot is significantly better than PN 1-shot, for both 1-shot and 5-shot evaluation settings, across datasets. Any thoughts on why that might be?

- In Table 2, in the “simple cross-entropy baselines” section, the Classifier-Baseline [5] and Baseline++ [2] should also be included.

Minor comments
-----------------------
- Line 55, in the query set definition, in the first tuple, the first label is y_1’. Should it just be y_1 (without the ‘)?
- Line  87 “the number of training distances that are lost” → “the number of training comparisons that are omitted”? The first phrase seems a little odd.
- When explaining the “Nearest Centroid” method (line 150): “This is the approach used by Prototypical Networks and SimpleShot”. It also is the approach used by Classifier-Baseline and Meta-Baseline [5].

References
----------------
- [1] Meta-learning with Latent Embedding Optimization. Rusu et al. ICLR 2019.
- [2] A closer look at few-shot classification. Chen et al. ICLR 2019.
- [3] Rethinking few-shot classification: A good embedding is all you need? Tian et al. 2020.
- [4] TADAM: Task dependent adaptive metric for improved few-shot learning. Oreshkin et al. NeurIPS 2018.
- [5] A New Meta-Baseline for Few-Shot Learning. Chen et al. 2020.
- [6] Fast and Flexible Multi-Task Classification Using Conditional Neural Adaptive Processes. Requeima et al. NeurIPS 2019.
- [7] Improved Few-Shot Visual Classification. Bateni et al. 2020
- [8] Learning a Universal Template for Few-shot Dataset Generalization. Triantafillou et al. ICML 2021.
- [9] A Simple Framework for Contrastive Learning of Visual Representations. Chen et al. ICML 2020.
- [10] A Theoretical Analysis of the Number of Shots in Few-shot Learning. Cao et al. ICLR 2020.
- [11] Model-Agnostic Meta-Learning for Fast Adaptation of Deep Networks. Finn et al. ICML 2017.
- [12] On First-order Meta-learning Algorithms. Nichol et al. 2018.
- [13] Few-shot Learning through an Information Retrieval Lens. Triantafillou et al. NeurIPS 2017.
- [14] How Important is the Train-Validation Split in Meta-Learning? Bai et al. 2021.
- [15] A Closer Look at the Training Strategy for Modern Meta-Learning. Chen et al. NeurIPS 2020.


**Time Spent Reviewing:**

4

---

> ### Author Response · Authors · 2021-08-10
> **Response to reviewer GQUE (1/2)**
>
> First of all we would like to thank reviewer GQUE for their time and detailed comments. It is great to hear that the reviewer found the paper enjoyable to read, and we appreciate that they recognized our formal and experimental demonstration of the inefficiency of episodic training. For clarity, we included a new reference list in part 2/2 of our comment, which we use throughout our response.
>
> ----
> **1.**
> > _“Perhaps 'scaling up' episodic methods [...] will result in surpassing the (analogously scaled up) NCA method, so in my view the jury is still out on this comparison.”_
>
> We believe that the evidence presented in the paper suggests that the trend found in Figure 3 should continue for batches of larger size. After all, as shown in Table 1, the extra number of gradients that the non-episodic approach can exploit wrt the episodic one grows quadratically with the batch size. What could happen is that the difference in performance between the two could converge to zero for very large batch sizes. We speculate this could be the case because the amount of “novelty” that new gradients provide when a large amount of training signal is already available is diminishing with the batch size. So it all depends on whether this amount of “novelty” decreases more or less rapidly than $O(w^2(m^2+n^2))$ (see Appendix A), which is something that also depends on the specific dataset in question. If it decreases more rapidly, then eventually for very large batch sizes non-episodic and episodic batches will converge to the same performance. Otherwise, the non-episodic approach will be increasingly better. In any case, we believe there is no evidence to believe that for larger batch sizes the episodic approach will at some point start outperforming the non-episodic one. Nonetheless, we thank the reviewer for raising this important point: we will add this discussion in the paper.
>
> As an extra data point, for tieredImageNet we train all of our methods on a batch size of 1024 (Appendix F), and we still find NCA performing better than MNs and PNs. Given our compute infrastructure, we are not able to test much larger batches, but we hope that the combinatorial argument given in the paper and extended above will suffice.
>
>
> ---
> **2.**
> > _“Can we theoretically or empirically study whether NCA-learned models are capable of performing well on tasks of a wide range of shots at test time, when using the nearest-centroid classification as the test-time algorithm?”_
>
> To answer to this question, we can consider the number of gradients lost by the episodic approach outlined in Table 1 and at the end of Appendix A, which is $w^2(w(m^2 + n^2 )−m− n)$ (if we consider both positive and negative pairs), where $w$ is the number of ways, and $n$ and $m$ the number of shots and queries respectively.
> Hence, we can say that if while increasing the number of shots we keep $w$ and $m$ fixed, then the advantage in terms of extra training signal will increase for the non-episodic NCA wrt to its episodic counterparts. However, one important caveat is that the _improvement_ in performance of the non-episodic variant wrt the non episodic one might saturate, because of what observed in the previous answer.
>
>
> ----
> **3.**
> > _”How should few-shot learning researchers change their approach [...]? PN/MN are no longer state-of-the-art models, and recent work these days is moving away from exclusively relying on episodic training for few-shot learning.”_
>
> In Section 1 of the paper we do observe that simple baselines and cross-entropy pre-training now generally outperform classic FSL meta-learning methods (see lines 29-36). The change in approach by FSL researchers has already happened: the goal of this paper is to understand the inefficiency of some of the most popular episodic methods to _retrospectively_ explain the recent abandonment of the strategy they pioneered. We opted to do so via a detailed case study of Matching and Prototypical Networks, which alongside MAML are perhaps the most influential approaches from the “new-wave” of meta-learning started circa 2016.
> Despite PNs/MNs not being state-of-the-art methods anymore, more recent methods (e.g. [A,B,C,D]) use them as building blocks. Moreover, several recent papers in applied machine learning do still make use of Prototypical Networks in real-world scenarios (e.g. Glaucoma grading [F], EEG analysis of Autism vs non-autism brain scans [G], and fault analysis of planetary gear boxes [H], just to name a few).
>
> ---
>
> **4.**
> >_“[try with] harder generalization problems to classes that are dissimilar to the training set (e.g. coming from different datasets)._
>
> We agree that testing NCA in the cross-domain setting is interesting, and we thank the reviewers for the suggestion. We performed experiments where models trained on miniImageNet are tested on CUB. We used the best hyper-parameters from Fig.1 in the Appendix for all models. On 1-shot/5-shot, NCA achieved 45.49/60.41, versus 44.62/59.75 for PNs and 44.88/58.51 for MNs (confidence intervals around ±0.15%). These results, which we will include in the paper, indicate that NCA maintains an advantage also in a cross-domain setting. These differences are slightly less pronounced than the same-domain setting, yet significant due to the narrow confidence intervals.
>
> ----
> **5.**
> > _”A potential advantage of PN is that we can modify its training (specifically, through the choice of the training shot hyperparameter)” [B] formally shows that the choice of the ‘shot’ hyperparameter during training impacts the properties of the learned embedding space [...].”_
>
> [B] is indeed relevant, and shows interesting theoretical results. However, their experiments differ from ours in a crucial aspect: we keep our batch sizes equal across training with different shots, whereas in [B] the increase in the number of shots (during training) also increases the batch size. Batch size is in general an important factor that affects deep neural networks training, and in cases like this, where it affects the combinatorics of the training pairs being exploitable, it becomes critical. More on this in the next answer.
>
> **6.**
> > _“In the original PN paper they showed empirically that it was beneficial to "match" the shot used at training and test times. The results here contradict this [...] Any thoughts on why that might be?”_
>
> In the original PNs paper, the batch sizes of 1-shot and 5-shot training configurations differ, while in ours they are kept constant (see lines 179-185 in our paper where we mention this). We believe that this explains the difference with our results.
> When batch sizes are equal, PNs’ 5-shot episodes generally exploit more pairs during training than 1-shot episodes, and we believe that this explains why our 5-shot models perform better than 1-shot models - on both 1 and 5-shot performance. Table 2 in Appendix E shows this effect clearly.
>
> ----
> **7.**
> > _“Is there a high-level take-away that connects the relationship of Reptile to MAML and that of PN (or MN) to NCA?”_
>
> This is an interesting question and something we touch on briefly in the related work section with respect to the work of Raghu et al. [E], who show that second order MAML updates mostly reuse features. [I] introduces Reptile as a first-order variant of MAML. The authors argue it works well because first-order updates using SGD implicitly provide a second-order MAML-like gradient update. This allows the model to generalise well between mini-batches/episodes, which is desirable in few-shot learning. They then hypothesise this being the reason why _fine-tuning_ (instead of meta-learning) works so well for few-shot learning.
> As a high-level takeaway with regards to our own analysis of MNs/PNS vs NCA: Non-episodic methods already allow for good generalisation to unseen classes - be it through improved batch exploitation (NCA) or implicit second-order gradient updates (Reptile), and omit the extra hyperparameters that episodic training introduce. By omitting the episodic structure, both these methods can better exploit the data and corresponding training signal that is available within a batch.
> ----
> **8.**
> > [Extra related work]
>
> Thank you for the references - we will include them in the related work section.
>
> [J] also show that the episodic strategy in meta-learning is inefficient, although for a different class of algorithms. They do this by providing both theoretical and experimental arguments on methods solving a convex optimization problem at the level of the base learner (such as MetaOptNet). Similar to us (though via a different analysis), they also show that the classic split (which in their paper they call _train-val_) is inefficient.
> [K] derives a generalisation bound for algorithms with a support/query separation. They do not provide any bounds for methods like NCA, which would be an interesting direction for future work.
> Finally, [L] too ignores the query/support separation in order to maximally exploit all the available samples while working in a Structured SVM framework (and optimizing for mean Average Precision). Though the reasoning about batch exploitation is analogous to ours, the scope of the paper is very different from ours.
>
> **[continues in response 2/2]**

---

> > ### Author Response · Authors · 2021-08-10
> > **Response to reviewer GQUE (2/2)**
> >
> > **9.**
> > > _”In the ablations, what is the PN model where prototypes are not computed? [...] Is this exactly the same as MN in this case? If so, I would just refer to it as MN.”_
> >
> > This is correct. That ablation indicates a method that uses the MNs approach while training, while using a nearest-centroid approach at test time (a-la prototypical networks). We do mention that this is the case on line 243, but we agree that calling it MNs in the figure would be clearer - we will adjust the paper accordingly.
> >
> > ----
> > **10.**
> > > _“In Table 2, in the 'simple cross-entropy baselines' section, the Classifier-Baseline [M] and Baseline++ [N] should also be included.”_
> >
> > Thank you for pointing this out - we will add them. Since both methods perform worse than the best results in our table in their respective domains, the overall take-home message won’t change.
> >
> > ----
> > **11.**
> >
> > Thanks also for the minor comments, we will update the paper accordingly.
> >
> > ----
> >
> > **References**
> >
> > * [A] Tadam: Task dependent adaptive metric for improved few-shot learning; Oreshkin et al.; NeurIPS 2018
> > * [B] A theoretical analysis of the number of shots in few-shot learning; Cao et al.; ICML 2020
> > * [C] Boosting few-shot visual learning with self-supervision; Gidaris et al.; CVPR 2019
> > * [D] Tapnet: Neural network augmented with task adaptive projection for few-shot learning; Yoon et al.; ICML 2019
> > * [E] Rapid learning or feature reuse? towards understanding the effectiveness of MAML; Raghu et al.;  ICLR 2020
> > * [F] Circumpapillary OCT-focused hybrid learning for glaucoma grading using tailored prototypical neural networks; Garcia et al.; Artificial Intelligence in Medicine, 2021
> > * [G] Understanding autism: the power of EEG harnessed by prototypical learning; Salekin et al.; In Proceedings of the Workshop on Medical Cyber Physical Systems and Internet of Medical Things, 2021
> > * [H] Multiscale dynamic fusion prototypical cluster network for fault diagnosis of planetary gearbox under few labeled samples; Li et al.; In Computers in Industry, 2020
> > * [I] On First-order Meta-learning Algorithms. Nichol et al. 2018.
> > * [J] How Important is the Train-Validation Split in Meta-Learning? Bai et al.; arXiv 2021.
> > * [K] A Closer Look at the Training Strategy for Modern Meta-Learning. Chen et al. NeurIPS 2020.
> > * [L] Few-Shot Learning Through an InformationRetrieval Lens; Triantafillou et al.; NeurIPS 2017
> > * [M] A New Meta-Baseline for Few-Shot Learning; Chen et al; 2020.
> > * [N] A closer look at few-shot classification. Chen et al. ICLR 2019.

---

> > > ### Comment · Reviewer_GQUE · 2021-08-15
> > > **response**
> > >
> > > Thank you for the additional interesting discussion and clarifications.
> > >
> > > Regarding point 2: this response discusses the additional training signal when increasing the number of shots, which to me relates to training efficiency. What I mentioned in my original review is something different which instead relates to shot robustness at test time. Specifically, for ProtoNets, even though large-shot training provides additional training signal compared to small-shot training, the latter is sometimes preferable, if evaluation will be performed on small-shot tasks. Specifically, by making the design choice of utilizing smaller training shots, the resulting embedding space has different properties (smaller intra-class variance) compared to large-shot training (which would prioritize larger inter-class variance instead), as shown in Cao et al. which is more appropriate for that setup. If instead using NCA, is there something analogous we can do at training time, in order to produce an embedding that works well for our desired setting of interest (e.g. small shot versus large shot tasks)? Or is NCA more shot robust in the sense that it does not require any training-time modifications to perform well in each setting of interest? I understand this question may be beyond the scope of the paper but I think it’s interesting to think about.
> > >
> > > More generally, I’m trying to understand, is it *always* preferable to use NCA over PN/MN? It seems to be that way if what we care about is training efficiency, and best exploiting the available training data (utilizing the largest number of comparisons within a batch). But are there other reasons (or specific cases) where episodic variants are preferable? There are certainly cases where episodic methods are valuable; as a simple example, non-episodic approaches do not directly offer a mechanism for implementing task conditioning which is very useful when dealing with heterogeneous task distributions. So it should be made clear wherever possible that the conclusions of this paper pertain only to the comparison between PN/MN and NCA and are not more general.
> > >
> > > Overall, I continue to think that analyzing the reason behind the inefficiency of PN/MN compared to NCA is interesting and a nice contribution. Although not the primary contribution of the paper, considering NCA as an additional strong baseline is also a nice contribution. The paper is really well written, accurately places the work in the relevant context, and performs thorough experimentation. A clear drawback of the paper in my opinion, however, is the narrow scope, and the potential limited impact since, as I mentioned in my original review, the field is already largely moving away from purely episodic models.

---

> > > > ### Author Response · Authors · 2021-08-20
> > > > **Additional response to reviewer GQUE**
> > > >
> > > > Thank you for the additional feedback and clarifications, and for engaging in the discussion.
> > > >
> > > > > _”is there something we can do at training time, in order to produce an embedding that works well for our desired setting of interest (e.g. small shot versus large shot tasks)?”_
> > > >
> > > >  > _”is NCA more shot robust in the sense that it does not require any training-time modifications to perform well in each setting of interest?”_
> > > >
> > > > What we show is that the NCA, _without_ any introduction nor tuning of setup-specific hyperparameters, outperforms PNs/MNs _with_ tuned setup-specific hyperparameters.
> > > > There is a subtle but important point to make here: this does not mean that an NCA-based method cannot benefit from a setup-specific choice of hyperparameters and achieve higher performance. After all, being able to tune three hyperparameters and then train a specific model per setup (like PNs/MNs do) should be a big advantage with respect to using a single model with fewer hyperparameters, no matter the method in consideration. To us, the fact that NCA _still_ outperforms PNs and MNs without having a similar experimental advantage is significant and illustrates how unnecessary the support/query separation is in the two methods analysed.
> > > >
> > > > Importantly, there is no technical limitation within the NCA loss that prevents it from being used together with sampling strategies that have several hyperparameters, which could be tuned across setup, or per-setup (if one is happy to further trade off simplicity for performance).
> > > > One could use a craftier sampling strategy for the NCA than the uniform random sampling baseline we use in the paper. For instance, in “Sampling Matter in Deep Embedding Learning” (ICCV 2017) Wu et al. draw samples uniformly according to their relative distance, which significantly stabilises training by diminishing the variance of gradients, and improves performance.
> > > >
> > > > Though investigating the choice of the sampling strategy (and its hyperparameters) is a promising direction and should be considered as future work, it is besides the point we make in this work.
> > > > Simply, our point is to show that the specific set of hyperparameters (#ways, #shots, #queries) inspired by the concept of support/query set separation (as expressed in Vinyals et al. seminal “Matching Networks” paper), is actually harmful _for MNs and PNs_. We decided to not tradeoff simplicity for performance because our objective wasn’t to glorify NCA, but rather discuss what we found out about support/query set separation in MNs/PNs and its associated hyperparameters.
> > > >
> > > >
> > > > > _I’m trying to understand, is it always preferable to use NCA over PN/MN?_
> > > >
> > > > In short yes, because NCA _before hyperparameter-tuning_ is pareto optimal with respect to PNs/MNs _after parameter tuning_. But see our previous answer for a more detailed argument.
> > > >
> > > > Thank you for bringing this up, we will hone the discussion in the paper to make this point clearer.
> > > >
> > > > > _“it should be made clear wherever possible that the conclusions of this paper pertain only to the comparison between PN/MN and NCA and are not more general._
> > > >
> > > > We agree it is important to be clear regarding the scope of the claims being made. We believe we specify that our scope is limited to MNs/PNs frequently (e.g. see line 10 in the Abstract and 37 in the Introduction).
> > > > However, we found two potentially misleading sentences: the sentences of line 86 and line 343 should be adjusted and made more specific to PNs and MNs. We will edit the text accordingly. Thanks for pointing this out and let us know if there are other potentially misleading passages.
> > > >
> > > > > _”A clear drawback of the paper in my opinion, however, is the narrow scope, and the potential limited impact since, as I mentioned in my original review, the field is already largely moving away from purely episodic models.”_
> > > >
> > > > We tried to address this specific concern in point 3 of our initial rebuttal comment, which we report and slightly edit below for convenience (note that the letters in squared brackets still refer to the referenced papers from our initial answer).
> > > >
> > > > We agree the field has already largely moved away from purely episodic models, and we mention this in our introduction by noting that simple baselines and cross-entropy pre-training are now outperforming classic FSL meta-learning methods and that are fairly standard choices.
> > > > This paper is still timely because it offers an angle to understand the inefficiency of some of the most popular episodic methods to retrospectively explain the recent abandonment of the strategy they pioneered. We opted to do so via a detailed case study of Matching and Prototypical Networks, which alongside MAML are perhaps the most influential approaches from the “new-wave” of meta-learning started circa 2016. Despite PNs/MNs not being state-of-the-art methods anymore, much more recent methods (e.g. [A,B,C,D]) use them as building blocks. Moreover, several recent papers in applied machine learning do still make use of Prototypical Networks in real-world scenarios (e.g. Glaucoma grading [F], EEG analysis of Autism vs non-autism brain scans [G], and fault analysis of planetary gear boxes [H], just to name a few).
> > > >
> > > > Finally, other papers that are very appreciated by the FSL community (see e.g. the analysis of Cao et al., or the one of Raghu et al.), too, focus their analysis on one or two methods only. Granted that the work is thorough, we believe that the narrow scope should not be considered as a negative quality, as it allows a better focused analysis.
> > > >
> > > > ----
> > > > We hope that the above clarifies our perspective and better addresses the concerns of this reviewer. Please do not hesitate to answer this message if further discussion is needed.

---

### Official Review · Reviewer_sPyu · 2021-07-25

**Rating:** 7
**Confidence:** 4

**Summary:**

This paper studies the question of whether episodes (using a split of support & query sets) are necessary for non-parameteric approaches for few-shot learning (examples include Prototypical Networks & Matching Networks). The authors propose a Neighborhood Component Analysis-based method for few-shot learning, where a mini-batch consists of examples from a subset of base classes, with no support or query split. The NCA loss then involves distances computed across all examples in the mini-batch rather than only using distances computed between support and query examples (as is done in Prototypical Networks & Matching Networks). The authors show that their proposed method is able to achieve better performance for 1-shot & 5-shot classification in miniImageNet, cifar-fs, and tieredImageNet benchmarks. They also speculate that the proposed method performs better because of its use of more distance computations in the loss compared to Prototypical Networks & Matching Networks and confirm this by conducting an experiments where they randomly drop distances in their NCA loss and showing this has a negative impact on performance.

**Ethical Concerns:**

No ethical concerns.

**Limitations And Societal Impact:**

Yes.

**Main Review:**

Originality: to my knowledge an NCA-type loss has not been considered for few-shot learning and the analysis comparing the differences between this loss and Prototypical Networks & Matching Networks loss is also original and interesting.

Quality: submission seems technically sound in terms of the analysis comparing their proposal to existing work. The experiments were also very useful, as they go beyond raw metrics and studied interested aspects of their method that show why it may be working better compared to previous work.

Clarity: the paper is very well-written - both description of the method(s) and the experiments are easy to understand and described well.

Significance: this work (along with other recent work showing that pre-training without any notion of episodes) could be very significant as they're questioning the benefit of more complicated episodic sampling-based approaches to few-shot learning. I do have some minor concerns/questions which could be clarified a bit further and help strengthen the paper.

1. It would be useful to include the results on metrics for Prototypical Networks on ResNet12 from previous work. The authors include results from their own implementation of Proto-Nets but I think including results from previous work is helpful to know that any implementation-specific choices by the authors in their recreation did not affect the metrics negatively.
2. Based on results of Figure 4, if NCA fixed-batch composition is always strictly better than the default NCA, why not treat fixed-batch composition as the default model? Is it because the default NCA is technically simpler since creating a batch just involves iterating through the data and the benefit only applies to small batches?

**Time Spent Reviewing:**

4

---

> ### Author Response · Authors · 2021-08-10
> **Response to reviewer sPyu**
>
> We thank reviewer sPyu for their time and feedback, and we are glad to know they have found our work original and interesting.
>
> Please note that, for convenience, in this comment we refer to papers with letters. The list of references is given at the end of the comment.
>
>
> ----
>
> > _“It would be useful to include the results on metrics for Prototypical Networks on ResNet12 from previous work.”_
>
> We agree reporting the results obtained by other Prototypical Networks re-implementations would be informative. We report comparisons below and we will add them to the supplementary material of the paper.
>
>
> | PNs implementation | Architecture | miniImageNet 1-shot | miniImageNet 5-shot |
> | --- | --- | ---| --- |
> | [A] | ResNet-10 | 51.98 | 72.64 |
> | [A] | ResNet-18 | 54.16 | 73.68 |
> | [A] | ResNet-34 | 53.90 | 74.65 |
> | [B] | WideResNet-28-10 | 55.85 | 68.72 |
> | [C] | ResNet-12 | 59.25 | 75.60 |
> | Ours (episodic setup from [D]) | ResNet-12 | 59.78 | 75.42 |
> | Ours (best episodic setup) | ResNet-12 | 61.32 | 77.77 |
>
>
> These results should corroborate what mentioned in section F of the supplementary material, i.e. that we made sure to pick good hyperparameters for all the methods we re-implemented, so that we can be sure of having competitive PNs and MNs baselines.
>
>
> ----
> > _“why not treat fixed-batch composition as the default model? Is it because the default NCA is technically simpler [..] ?”_
>
>
> Given that the fixed-batch NCA variant is only very marginally outperforming the random-batches variant, we indeed opted for using the latter for the sake of simplicity. Using a fixed batch composition would require the introduction and tuning of more hyperparameters, which partially defeats the spirit of our paper.
>
>
> However, the composition of examples populating the batch in terms of intra-class and inter-class (semantic) distances should be an interesting subject, as it could lead to effective ways to populate batches during FSL training.
> For instance: it could be useful to adopt a curriculum learning approach in which earlier-encountered batches only require to discriminate between classes very different from each other in terms of semantic, and then gradually present more fine-grained examples while training progresses.
>
>
> Another way to go beyond random sampling as a strategy to determine batch composition is to adopt the approach proposed in [E] for deep metric learning, in which each element in a batch is sampled according to their relative distance from one another, which greatly helps to reduce the variance of the gradients and stabilise training.
>
> To summarise: though beyond the scope of this paper, we believe that variations on the batch-composition strategy point in the direction of promising future work.
>
> ----
>
> **References**
> * [A] A Closer Look at Few-shot Classification; Chen et al.; ICLR 2019
> * [B] Boosting Few-Shot Visual Learning with Self-Supervision; Gidaris et al.; ICCV 2019
> * [C] Meta-Learning with Differentiable Convex Optimization; Lee et al.; CVPR 2019
> * [D] Prototypical Networks for Few-shot Learning; Snell et al.; NeurIPS 2017
> * [E] Sampling Matters in Deep Embedding Learning; Wu et al.; ICCV 2017

---

> > ### Comment · Reviewer_sPyu · 2021-09-01
> > **Thanks for the response**
> >
> > I thank the authors for their response and for addressing the specific questions in my review - this was very useful.

---

### Decision · Program_Chairs · 2021-09-27

**Decision:**

Accept (Poster)

**Comment:**

The paper is currently very borderline with respect to reviewer opinions. The paper is well-written, with a straightforward premise and thorough experiments. The main outstanding concern is the narrow scoping: PN/MN are no longer considered the state of the art in the field, and the narrow focus of the paper makes it unclear how well the findings will generalize to other forms of episodic learning. On the other hand, these are foundational models and are still used in many contexts. The heart of the debate is whether this will help move the field forward. I think that the NCA baseline is simple and effective enough that the answer leans towards “yes.” In particular, if this was published several years ago, I think it is likely that NCA itself could have become a foundational approach to few-shot learning - especially since it requires less tuning.

I think that the relationship between performance, and the number of pairwise distances per batch is compelling. I would like to see the following experiment (which I didn’t find in the current paper): take PN and/or MN and for each episode, randomize the episode several times. That is, take the set of support and query points, compute the loss, shuffle the batch (assigning different query/support points within the batch), and then compute the loss again. Take the loss of the batch to be the average of these, and then take a gradient step. This effectively increases the number of comparisons used in the batch, but in a way that PN/MN can exploit. Based on the hypothesis in the paper, I expect the PN/MN results to improve, but it will be interesting either way.